# Uncertainty Estimation in Autoregressive Structured Prediction

**Andrey Malinin**
Yandex, Higher School of Economics
am969@yandex-team.ru

**Mark Gales**
ALTA Institute, University of Cambridge
mjfg@eng.cam.ac.uk

## Abstract

Uncertainty estimation is important for ensuring safety and robustness of AI systems. While most research in the area has focused on unstructured prediction tasks, limited work has investigated general uncertainty estimation approaches for structured prediction. Thus, this work aims to investigate uncertainty estimation for autoregressive structured prediction tasks within a single unified and interpretable probabilistic ensemble-based framework. We consider uncertainty estimation for sequence data at the token-level and complete sequence-level; interpretations for, and applications of, various measures of uncertainty; and discuss both the theoretical and practical challenges associated with obtaining them. This work also provides baselines for token-level and sequence-level error detection, and sequence-level out-of-domain input detection on the WMT'14 English-French and WMT'17 English-German translation and LibriSpeech speech recognition datasets.

## 1 Introduction

Neural Networks (NNs) have become the dominant approach in numerous applications (Simonyan & Zisserman, 2015; Mikolov et al., 2013; 2010; Bahdanau et al., 2015; Vaswani et al., 2017; Hinton et al., 2012) and are being widely deployed in production. As a consequence, predictive uncertainty estimation is becoming an increasingly important research area, as it enables improved safety in automated decision making (Amodei et al., 2016). Important advancements have been the definition of baseline tasks and metrics (Hendrycks & Gimpel, 2016) and the development of ensemble approaches, such as Monte-Carlo Dropout (Gal & Ghahramani, 2016) and Deep Ensembles (Lakshminarayanan et al., 2017)[1]. Ensemble-based uncertainty estimates have been successfully applied to detecting misclassifications, out-of-distribution inputs and adversarial attacks (Carlini & Wagner, 2017; Smith & Gal, 2018; Malinin & Gales, 2019) and to active learning (Kirsch et al., 2019). Crucially, they allow *total uncertainty* to be decomposed into *data uncertainty*, the intrinsic uncertainty associated with the task, and *knowledge uncertainty*, which is the model's uncertainty in the prediction due to a lack of understanding of the data (Malinin, 2019)[2]. Estimates of *knowledge uncertainty* are particularly useful for detecting anomalous and unfamiliar inputs (Kirsch et al., 2019; Smith & Gal, 2018; Malinin & Gales, 2019; Malinin, 2019).

Despite recent advances, most work on uncertainty estimation has focused on unstructured tasks, such as image classification. Meanwhile, uncertainty estimation within a general, *unsupervised*, probabilistically interpretable ensemble-based framework for structured prediction tasks, such as language modelling, machine translation (MT) and speech recognition (ASR), has received little attention. Previous work has examined bespoke *supervised* confidence estimation techniques for each task separately (Evermann & Woodland, 2000; Liao & Gales, 2007; Ragni et al., 2018; Chen et al., 2017; Koehn, 2009; Kumar & Sarawagi, 2019) which construct an "error-detection" model on top of the original ASR/NMT system. While useful, these approaches suffer from a range of limitations. Firstly, they require a *token-level* supervision, typically obtained via minimum edit-distance alignment to a ground-truth transcription (ASR) or translation (NMT), which can itself by noisy. Secondly, such token-level supervision is generally inappropriate for translation, as it doesn't account for the validity of re-arrangements. Thirdly, we are unable to determine whether the error is due to *knowledge* or

---

[1] An in-depth comparison of ensemble methods was conducted in (Ashukha et al., 2020; Ovadia et al., 2019)

[2] Data and Knowledge Uncertainty are sometimes also called Aleatoric and Epistemic uncertainty.

*data uncertainty*. Finally, this model is itself subject to the pitfalls of the original system - domain shift, noise, etc. Thus, *unsupervised* uncertainty-estimation methods are more desirable.

Recently, however, initial investigations into *unsupervised* uncertainty estimation for structured prediction have appeared. The nature of *data uncertainty* for translation tasks was examined in (Ott et al., 2018a). Estimation of sequence and word-level uncertainty estimates via Monte-Carlo Dropout ensembles has been investigated for machine translation (Xiao et al., 2019; Wang et al., 2019; Fomicheva et al., 2020). However, these works focus on machine translation, consider only a small range of uncertainty adhoc measures, provide limited theoretical analysis of their properties and do not make explicit their limitations. Furthermore, they don't identify or tackle challenges in estimating uncertainty arising from exponentially large output space. Finally, to our knowledge, no work has examined uncertainty estimation for autoregressive ASR models.

This work examines uncertainty estimation for structured prediction tasks within a general, probabilistically interpretable ensemble-based framework. The five core contributions are as follows. First, we derive information-theoretic measures of both *total uncertainty* and *knowledge uncertainty* at both the *token level* and the *sequence level*, make explicit the challenges involved and state any assumptions made. Secondly, we introduce a novel uncertainty measure, *reverse mutual information*, which has a set of desirable attributes for structured uncertainty. Third, we examine a range of Monte-Carlo approximations for sequence-level uncertainty. Fourth, for structured tasks there is a choice of how ensembles of models can be combined; we examine how this choice impacts predictive performance and derived uncertainty measures. Fifth, we explore the practical challenges associated with obtaining uncertainty estimates for structured predictions tasks and provide performance baselines for token-level and sequence-level error detection, and out-of-domain (OOD) input detection on the WMT'14 English-French and WMT'17 English-German translation datasets and the LibriSpeech ASR dataset.

## 2 Uncertainty for Structured Prediction

In this section we develop an ensemble-based uncertainty estimation framework for structured prediction and introduce a novel uncertainty measure. We take a Bayesian viewpoint on ensembles, as it yields an elegant probabilistic framework within which interpretable uncertainty estimates can be obtained. The core of the Bayesian approach is to treat the model parameters $\boldsymbol{\theta}$ as random variables and place a prior $p(\boldsymbol{\theta})$ over them to compute a posterior $p(\boldsymbol{\theta}|\mathcal{D})$ via Bayes' rule, where $\mathcal{D}$ is the training data. Unfortunately, exact Bayesian inference is intractable for neural networks and it is necessary to consider an explicit or implicit approximation $q(\boldsymbol{\theta})$ to the true posterior $p(\boldsymbol{\theta}|\mathcal{D})$ to generate an ensemble. A number of different approaches to generating ensembles have been developed, such as Monte-Carlo Dropout (Gal & Ghahramani, 2016) and DeepEnsembles (Lakshminarayanan et al., 2017). An overview is available in (Ashukha et al., 2020; Ovadia et al., 2019).

Consider an ensemble of models $\{P(\boldsymbol{y}|\boldsymbol{x};\boldsymbol{\theta}^{(m)})\}_{m=1}^M$ sampled from an approximate posterior $q(\boldsymbol{\theta})$, where each model captures the mapping between variable-length sequences of inputs $\{x_1,\cdots,x_T\} = \boldsymbol{x} \in \mathcal{X}$ and targets $\{y_1,\cdots,y_L\} = \boldsymbol{y} \in \mathcal{Y}$, where $x_t \in \{w_1,\cdots,w_V\}$, $y_l \in \{\omega_1,\cdots,\omega_K\}$. The *predictive posterior* is obtained by taking the expectation over the ensemble:

$$P(\boldsymbol{y}|\boldsymbol{x},\mathcal{D}) = \mathbb{E}_{q(\boldsymbol{\theta})}\big[P(\boldsymbol{y}|\boldsymbol{x},\boldsymbol{\theta})\big] \approx \frac{1}{M}\sum_{m=1}^M P(\boldsymbol{y}|\boldsymbol{x},\boldsymbol{\theta}^{(m)}), \ \boldsymbol{\theta}^{(m)} \sim q(\boldsymbol{\theta}) \approx p(\boldsymbol{\theta}|\mathcal{D}) \qquad (1)$$

The *total uncertainty* in the prediction of $\boldsymbol{y}$ is given by the *entropy* of the predictive posterior.

$$\underbrace{\mathcal{H}[P(\boldsymbol{y}|\boldsymbol{x},\mathcal{D})]}_{\text{Total Uncertainty}} = \mathbb{E}_{P(\boldsymbol{y}|\boldsymbol{x},\mathcal{D})}[-\ln P(\boldsymbol{y}|\boldsymbol{x},\mathcal{D})] = -\sum_{\boldsymbol{y}\in\mathcal{Y}} P(\boldsymbol{y}|\boldsymbol{x},\mathcal{D})\ln P(\boldsymbol{y}|\boldsymbol{x},\mathcal{D}) \qquad (2)$$

The sources of uncertainty can be decomposed via the *mutual information* $\mathcal{I}$ between $\boldsymbol{\theta}$ and $\boldsymbol{y}$:

$$\underbrace{\mathcal{I}\big[\boldsymbol{y},\boldsymbol{\theta}|\boldsymbol{x},\mathcal{D}\big]}_{\text{Know. Uncertainty}} = \mathbb{E}_{q(\boldsymbol{\theta})}\Big[\mathbb{E}_{P(\boldsymbol{y}|\boldsymbol{x},\boldsymbol{\theta})}\Big[\ln\frac{P(\boldsymbol{y}|\boldsymbol{x},\boldsymbol{\theta})}{P(\boldsymbol{y}|\boldsymbol{x},\mathcal{D})}\Big]\Big] = \underbrace{\hat{\mathcal{H}}\big[P(\boldsymbol{y}|\boldsymbol{x},\mathcal{D})\big]}_{\text{Total Uncertainty}} - \underbrace{\mathbb{E}_{q(\boldsymbol{\theta})}\big[\hat{\mathcal{H}}[P(\boldsymbol{y}|\boldsymbol{x},\boldsymbol{\theta})]\big]}_{\text{Expected Data Uncertainty}} \qquad (3)$$

Mutual information (MI) is a measure of 'disagreement' between models in the ensemble, and therefore a measure of *knowledge uncertainty* (Malinin, 2019). It can be expressed as the difference between the entropy of the predictive posterior and the expected entropy of each model in the ensemble.

The former is a measure of *total uncertainty* and the latter is a measure of *data uncertainty* (Depeweg et al., 2017). Another measure of ensemble diversity is the expected pairwise KL-divergence (EPKL):

$$\mathcal{K}\big[\boldsymbol{y}, \boldsymbol{\theta}|\boldsymbol{x}, \mathcal{D}\big] = \mathbb{E}_{\mathrm{q}(\boldsymbol{\theta})\mathrm{q}(\tilde{\boldsymbol{\theta}})}\Big[\mathbb{E}_{\mathrm{P}(\boldsymbol{y}|\boldsymbol{x}, \boldsymbol{\theta})}\Big[\ln\frac{\mathrm{P}(\boldsymbol{y}|\boldsymbol{x}, \boldsymbol{\theta})}{\mathrm{P}(\boldsymbol{y}|\boldsymbol{x}, \tilde{\boldsymbol{\theta}})}\Big]\Big] \quad \mathrm{q}(\boldsymbol{\theta}) \approx \mathrm{p}(\boldsymbol{\theta}|\mathcal{D}) \tag{4}$$

where $\mathrm{q}(\boldsymbol{\theta}) = \mathrm{q}(\tilde{\boldsymbol{\theta}})$ and $\tilde{\boldsymbol{\theta}}$ is a dummy variable. This measure is an upper bound on the mutual information, obtainable via Jensen's inequality. A novel measure of diversity which we introduce in this work is the *reverse mutual information* (RMI) between each model and the predictive posterior:

$$\mathcal{M}\big[\boldsymbol{y}, \boldsymbol{\theta}|\boldsymbol{x}, \mathcal{D}\big] = \mathbb{E}_{\mathrm{q}(\boldsymbol{\theta})}\Big[\mathbb{E}_{\mathrm{P}(\boldsymbol{y}|\boldsymbol{x}, \mathcal{D})}\Big[\ln\frac{\mathrm{P}(\boldsymbol{y}|\boldsymbol{x}, \mathcal{D})}{\mathrm{P}(\boldsymbol{y}|\boldsymbol{x}, \boldsymbol{\theta})}\Big]\Big], \quad \mathrm{q}(\boldsymbol{\theta}) \approx \mathrm{p}(\boldsymbol{\theta}|\mathcal{D}) \tag{5}$$

This is the reverse-KL divergence counterpart to the mutual information (3), and has not been previously explored. As will be shown in the next section, RMI is particularly attractive for estimating uncertainty in structured prediction. Interestingly, RMI is the difference between EPKL and MI:

$$\mathcal{M}\big[\boldsymbol{y}, \boldsymbol{\theta}|\boldsymbol{x}, \mathcal{D}\big] = \mathcal{K}\big[\boldsymbol{y}, \boldsymbol{\theta}|\boldsymbol{x}, \mathcal{D}\big] - \mathcal{I}\big[\boldsymbol{y}, \boldsymbol{\theta}|\boldsymbol{x}, \mathcal{D}\big] \geq 0 \tag{6}$$

While mutual information, EPKL and RMI yield estimates of *knowledge uncertainty*, only mutual information 'cleanly' decomposes into *total* and *data uncertainty*. EPKL and RMI *do not* yield clean measures of *total* and *data uncertainty*, respectively. For details see appendix A.

Unfortunately, we cannot in practice construct a model which directly yields a distribution over an infinite set of variable-length sequences $\boldsymbol{y} \in \mathcal{Y}$. Neither can we take expectations over the this set. Instead, autoregressive models are used to factorize the joint distribution over $\boldsymbol{y}$ into a product of conditionals over a finite set of classes, such as words or BPE tokens (Sennrich et al., 2015).

$$\mathrm{P}(\boldsymbol{y}|\boldsymbol{x}, \boldsymbol{\theta}) = \prod_{l=1}^{L}\mathrm{P}(y_l|\boldsymbol{y}_{<l}, \boldsymbol{x}; \boldsymbol{\theta}), \quad x_t \in \{w_1, \cdots, w_V\}, \; y_l \in \{\omega_1, \cdots, \omega_K\} \tag{7}$$

Here the distribution over each $y_l$ is conditioned on all the previous $\boldsymbol{y}_{<l} = \{y_1, \cdots, y_{l-1}\}$, which we shall refer to as the *context*. This set of conditional independence assumptions allows us to define a model on the *finite* space $\mathcal{Y}^L$. This formulation describes all machine translation (Bahdanau et al., 2015; Vaswani et al., 2017), end-to-end speech recognition (Chan et al., 2015) and other related tasks. Alternative factorization orders, representing different conditional independence assumptions, can be considered. However, without loss of generality, we will use the standard factorization.

For these models uncertainty estimation can be examined at two levels - the *token level*, which considers uncertainty in the prediction of a single $y_l$, and the *sequence level*, which considers the uncertainty of predicting the entire sequence $\boldsymbol{y}$. *Token-level* uncertainty estimation for autoregressive models is isomorphic to un-structured uncertainty estimation with additional conditioning on the context $\boldsymbol{y}_{<l}$, and so is presented in appendix A.1. Instead, we focus on discussing *sequence-level* uncertainty estimation in autoregressive models. We have chosen to focus on autoregressive models as they present interesting challenges, are more general than models with stronger conditional independence assumptions, and are widely applied to tasks of practical value. We emphasize that the proposed ensemble-based approach is general and can be applied to structured tasks with different conditional independence assumptions (Graves et al., 2006; Gales & Young, 2008; Gu et al., 2017)

## 3 MONTE-CARLO APPROXIMATIONS

The key challenge of autoregressive models is that expressions (2)-(5) are intractable to evaluate. Specifically, all expectations over $\boldsymbol{y}$ are intractable to evaluate due to the combinatorial explosion of the hypothesis space - there are a total of $|K|^L$ possible L-length sequences in $\mathcal{Y}^L$, where $K$ is the vocabulary size, and it is necessary to do a forward-pass through the model *for each* hypothesis. This is an issue which was ignored in prior work (Wang et al., 2019; Fomicheva et al., 2020; Xiao et al., 2019). Clearly, it is necessary to consider Monte-Carlo approximations to make this tractable. A key desiderata of these approximations is that they should be obtainable *at no extra cost* on top of standard beam-search inference from the ensemble. We examine two types of Monte-Carlo (MC) approximations for expressions (2)-(5) which are identical in the limit, but have different attributes given a finite sample size. Properties of these approximation are detailed in appendix A.

A result of the auto-regressive conditional independence assumption is that distributions over long sequences can have higher entropy than over short ones. To compare uncertainties of sequences of different lengths, in accordance with the desiderata in the introduction, we consider length-normalized '*rate*' (Cover & Thomas, 2006) equivalents of all uncertainty measures, denoted by '∧'.

The simplest Monte-Carlo estimation for entropy is to approximate (2) using $S$ samples:

$$\hat{\mathcal{H}}_{\text{S-MC}}^{(S)}\big[\text{P}(\boldsymbol{y}|\boldsymbol{x},\mathcal{D})\big] \approx -\frac{1}{S}\sum_{s=1}^{S}\frac{1}{L^{(s)}}\ln\text{P}(\boldsymbol{y}^{(s)}|\boldsymbol{x},\mathcal{D}),\ \boldsymbol{y}^{(s)} \sim \text{P}(\boldsymbol{y}|\boldsymbol{x},\mathcal{D}) \tag{8}$$

where $\boldsymbol{y}^{(s)}$ is a realization of the random variable $\boldsymbol{y}$. Alternatively, we can approximate (2) as a sum of conditional entropies via the entropy chain-rule (Cover & Thomas, 2006):

$$\hat{\mathcal{H}}_{\text{C-MC}}^{(S)}[\text{P}(\boldsymbol{y}|\boldsymbol{x},\mathcal{D})] \approx \frac{1}{S}\sum_{s=1}^{S}\frac{1}{L^{(s)}}\sum_{l=1}^{L^{(s)}}\mathcal{H}[\text{P}(y_l|\boldsymbol{y}_{<l}^{(s)},\boldsymbol{x},\mathcal{D})],\quad \forall\boldsymbol{y}_{<l}^{(s)} \subset \boldsymbol{y}^{(s)} \sim \text{P}(\boldsymbol{y}|\boldsymbol{x},\mathcal{D}) \tag{9}$$

Given a set of consistent contexts $\boldsymbol{y}_{<l} \subset \boldsymbol{y}\ \forall l \leq L$, this approximation reduces to averages of token-level uncertainty estimates as a consequence of the entropy chain-rule.

Approximations (8) and (9) both yield exact estimates of *total uncertainty* (2) in the limit as $S \to \infty$. However, (8) only considers the probabilities of individual tokens $y_l^{(s)}$ along a hypothesis $\boldsymbol{y}^{(s)}$, while (9) considers the entire conditional distribution over each $y_l$. Consequently, (9) may yield a more stable approximation using a smaller number of samples. At the same time, while (8) yields a noisier estimate for a finite $S$, it is more sensitive to the *particular set of hypotheses considered*.

Monte-Carlo approximations can also be considered for mutual information (3):

$$\hat{\mathcal{I}}_{\text{MC}}^{(S)}\big[\boldsymbol{y},\boldsymbol{\theta}|\boldsymbol{x},\mathcal{D}\big] \approx \frac{1}{S}\sum_{s=1}^{S}\frac{1}{L^{(s)}}\ln\frac{\text{P}(\boldsymbol{y}^{(s)}|\boldsymbol{x},\boldsymbol{\theta}^{(s)})}{\text{P}(\boldsymbol{y}^{(s)}|\boldsymbol{x},\mathcal{D})},\quad \boldsymbol{y}^{(s)} \sim \text{P}(\boldsymbol{y}|\boldsymbol{X},\boldsymbol{\theta}^{(s)}),\ \boldsymbol{\theta}^{(s)} \sim \text{q}(\boldsymbol{\theta}) \tag{10}$$

Unfortunately, this requires sampling from *each* model individually - obtaining this estimate does not come 'for free' with standard ensemble inference. An efficient approximation can be obtained via the relative-entropy chain-rule (Cover & Thomas, 2006):

$$\hat{\mathcal{I}}_{\text{C-MC}}^{(S)}\big[\boldsymbol{y},\boldsymbol{\theta}|\boldsymbol{x},\mathcal{D}\big] \approx \frac{1}{S}\sum_{s=1}^{S}\frac{1}{L^{(s)}}\sum_{l=1}^{L^{(s)}}\mathcal{I}[y_l,\boldsymbol{\theta}|\boldsymbol{y}_{<l}^{(s)},\boldsymbol{x},\mathcal{D}],\quad \forall\boldsymbol{y}_{<l}^{(s)} \subset \boldsymbol{y}^{(s)} \sim \text{P}(\boldsymbol{y}|\boldsymbol{x},\mathcal{D}) \tag{11}$$

Unlike (10), (11) uses samples from the predictive posterior and reduces to averages of token-level mutual information $\mathcal{I}[y_l,\boldsymbol{\theta}|\boldsymbol{y}_{<l}^{(s)},\boldsymbol{x},\mathcal{D}]$. Thus, it is obtained at *no extra cost* with standard ensemble inference. However, it will *not* yield (3) as $S \to \infty$. Nevertheless, this estimate may still be useful in practice. An approximation $\hat{\mathcal{K}}_{\text{C-MC}}^{(S)}$ for EPKL (4) with identical properties is described in appendix A.

Asymptotically exact joint-sequence and chain-rule MC estimates of *knowledge uncertainty can* be obtained 'for free' during inference from $\text{P}(\boldsymbol{y}|\boldsymbol{x},\mathcal{D})$ by considering the new measure RMI (5):

$$\hat{\mathcal{M}}_{\text{S-MC}}^{(S)}\big[\boldsymbol{y},\boldsymbol{\theta}|\boldsymbol{x},\mathcal{D}\big] \approx \mathbb{E}_{\text{q}(\boldsymbol{\theta})}\Big[\frac{1}{S}\sum_{s=1}^{S}\frac{1}{L^{(s)}}\ln\frac{\text{P}(\boldsymbol{y}^{(s)}|\boldsymbol{x},\mathcal{D})}{\text{P}(\boldsymbol{y}^{(s)}|\boldsymbol{x},\boldsymbol{\theta})}\Big],\quad \boldsymbol{y}^{(s)} \sim \text{P}(\boldsymbol{y}|\boldsymbol{x},\mathcal{D}) \tag{12}$$

$$\hat{\mathcal{M}}_{\text{C-MC}}^{(S)}\big[\boldsymbol{y},\boldsymbol{\theta}|\boldsymbol{x},\mathcal{D}\big] \approx \frac{1}{S}\sum_{s=1}^{S}\frac{1}{L^{(s)}}\sum_{l=1}^{L^{(s)}}\mathcal{M}[y_l,\boldsymbol{\theta}|\boldsymbol{y}_{<l}^{(s)},\boldsymbol{x},\mathcal{D}],\quad \forall\boldsymbol{y}_{<l}^{(s)} \subset \boldsymbol{y}^{(s)} \sim \text{P}(\boldsymbol{y}|\boldsymbol{x},\mathcal{D}) \tag{13}$$

Similar to (9) and (11), (13) is also an average of token-level RMI $\mathcal{M}[y_l,\boldsymbol{\theta}|\boldsymbol{y}_{<l}^{(s)},\boldsymbol{x},\mathcal{D}]$, and like (8), (12) is sensitive to a particular set of hypotheses, while (13) is more stable.

**Practical Considerations** Before applying the proposed Monte-Carlo approximations, two practicalities need to be considered. Firstly, due to the vastness of the hypothesis space $\mathcal{Y}^L$, Monte-Carlo sampling requires prohibitively many samples and the introduction of a decision rule, whose approximation may require a costly computation and whose effectiveness is not yet fully understood, to find a good set of hypotheses (Eikema & Aziz, 2020; Holtzman et al., 2019). Instead, beam-search

is typically used for inference, as it efficiently finds high-quality hypotheses. With regards to the Monte-Carlo estimators above, beam-search can be interpreted as a form of importance-sampling which yields hypotheses from high-probability regions of the hypothesis space. As each hypothesis is seen only *once* during beam-search, the uncertainty associated with each hypothesis $\boldsymbol{y}^{(b)}$ within a beam $\mathcal{B}$ in the MC estimators above must be importance-weighted in proportion to $\mathrm{P}(\boldsymbol{y}^{(b)}|\boldsymbol{x}, \mathcal{D})$.

$$\hat{\mathcal{H}}_{\text{S-IW}}^{(B)}[\mathrm{P}(\boldsymbol{y}|\boldsymbol{x}, \mathcal{D})] \approx -\sum_{b=1}^{B} \frac{\pi_b}{L^{(b)}} \ln \mathrm{P}(\boldsymbol{y}^{(b)}|\boldsymbol{x}, \mathcal{D}), \boldsymbol{y}^{(b)} \in \mathcal{B}, \pi_b = \frac{\exp \frac{1}{T} \ln \mathrm{P}(\boldsymbol{y}^{(b)}|\boldsymbol{x}, \mathcal{D})}{\sum_k^B \exp \frac{1}{T} \ln \mathrm{P}(\boldsymbol{y}^{(k)}|\boldsymbol{x}, \mathcal{D})} \quad (14)$$

$$\hat{\mathcal{H}}_{\text{C-IW}}^{(B)}[\mathrm{P}(\boldsymbol{y}|\boldsymbol{x}, \mathcal{D})] \approx \sum_{b=1}^{B} \sum_{l=1}^{L^{(b)}} \frac{\pi_b}{L^{(b)}} \mathcal{H}[\mathrm{P}(y_l|\boldsymbol{x}, \boldsymbol{y}_{<l}^{(b)}, \mathcal{D})] \quad (15)$$

Note that here we introduce *temperature calibration $T$*, which allows us to 'soft-adjust' the contribution of the lower-probability hypotheses to the resulting measures of uncertainty. Higher temperature make the importance weights more uniform across the hypotheses. The effects of this are detailed in appendix G. Equivalent expressions for $\hat{\mathcal{I}}_{\text{C-IW}}^{(B)}$, $\hat{\mathcal{K}}_{\text{C-IW}}^{(B)}$, $\hat{\mathcal{M}}_{\text{S-IW}}^{(B)}$ and $\hat{\mathcal{M}}_{\text{C-IW}}^{(B)}$ are provided in appendix A.

Second, we must consider how to obtain the predictive posterior for an ensemble of autoregressive models. The models can be combined either as a *expectation-of-products* or a *product-of-expectations*:

$$\mathrm{P}_{\text{EP}}(\boldsymbol{y}|\boldsymbol{x}, \mathcal{D}) = \mathbb{E}_{\mathrm{q}(\boldsymbol{\theta})}\Big[\prod_{l=1}^{L} \mathrm{P}(y_l|\boldsymbol{y}_{<l}, \boldsymbol{x}, \boldsymbol{\theta})\Big], \ \mathrm{P}_{\text{PE}}(\boldsymbol{y}|\boldsymbol{x}, \mathcal{D}) = \prod_{l=1}^{L} \mathbb{E}_{\mathrm{q}(\boldsymbol{\theta})}\Big[\mathrm{P}(y_l|\boldsymbol{y}_{<l}, \boldsymbol{x}, \boldsymbol{\theta})\Big] \quad (16)$$

The former represents *sequence-level Bayesian model averaging*, while the latter *token-level Bayesian model averaging*. Both are methods to do model combination[3], but only the former is fully consistent with the sequence-level uncertainty measure defined in section 2, as they assume that all tokens in the sequence $\boldsymbol{y}$ are generated from the same $\boldsymbol{\theta}^{(m)}$. However, it is not clear a-priori which combination yields superior predictive performance given a set of samples of parameters $\boldsymbol{\theta}^{(m)}$ from a particular $\mathrm{q}(\boldsymbol{\theta})$ and an inference method. If hypotheses are obtained via beam-search, which is typically a sequence of token-level decisions, considering $\mathrm{P}_{\text{PE}}(\boldsymbol{y}|\boldsymbol{x}, \mathcal{D})$ *may* be advantageous. The choice of combination also affects how the token-level predictive posterior $\mathrm{P}(y_l|\boldsymbol{y}_{<l}^{(b)}, \boldsymbol{x}, \mathcal{D})$ is obtained:

$$\mathrm{P}_{\text{EP}}(y_l|\boldsymbol{y}_{<l}^{(b)}, \boldsymbol{x}, \mathcal{D}) = \frac{\mathbb{E}_{\mathrm{q}(\boldsymbol{\theta})}[\mathrm{P}(y_l, \boldsymbol{y}_{<l}^{(b)}, \boldsymbol{x}, \boldsymbol{\theta})]}{\mathbb{E}_{\mathrm{q}(\boldsymbol{\theta})}[\mathrm{P}(\boldsymbol{y}_{<l}^{(b)}, \boldsymbol{x}, \boldsymbol{\theta})]}, \ \mathrm{P}_{\text{PE}}(y_l|\boldsymbol{y}_{<l}^{(b)}, \boldsymbol{x}, \mathcal{D}) = \mathbb{E}_{\mathrm{q}(\boldsymbol{\theta})}[\mathrm{P}(y_l|\boldsymbol{y}_{<l}^{(b)}, \boldsymbol{x}, \boldsymbol{\theta})] \quad (17)$$

This choice affects measures derived from the sequence and token-level predictive posteriors. However, all measures can still be calculated for *both* forms at the same time, regardless of which was used for inference. Thus, the choice of combination depends on which yields superior performance.

## 4 EXPERIMENTAL EVALUATION

The current section provides performance baselines on three applications of structured uncertainty estimates: sequence-level and token-level error detection, and out-of-distribution input (anomaly) detection. Additional analysis is provided in appendices C-J. We also compare performance to prior heuristic ensemble-based approaches. This work only considers ensembles of autoregressive neural machine translation (NMT) and speech recognition (ASR) models generated by training identical models from different random initializations (Lakshminarayanan et al., 2017). This approach was shown to consistently outperform other ensemble generation techniques using exponentially smaller ensembles (Ashukha et al., 2020; Ovadia et al., 2019; Fort et al., 2019). Ensembles of 10 transformer-big (Vaswani et al., 2017) models were trained on the WMT'17 English-to-German (EN-DE) and WMT'14 English-to-French (EN-FR) translation tasks and evaluated on the newstest14 (nwt14) dataset. All models were trained using the configuration described in (Ott et al., 2018b). Ensembles of 6 VGG-Transformer (Mohamed et al., 2019) models were trained on the LibriSpeech (Panayotov et al., 2015) (LSP) ASR dataset. Standard Fairseq (Ott et al., 2019) implementations of all models are used. Details of model configurations are available in appendix B. Note that no comparison is made

---

[3]In the current Fairseq (Ott et al., 2019) implementation ensembles are combined as a *product-of-expectations*.

to *supervised* uncertainty estimation techniques for NMT/ASR, such as those described in (Liao & Gales, 2007; Koehn, 2009), for two reasons. Firstly, the focus of this work is general, *unsupervised* uncertainty estimation approaches based on ensemble methods. Secondly, to our knowledge, they have not been applied to autoregressive models and doing so is beyond the scope of this work.

Table 1: Predictive performance in terms of BLEU, %WER and NLL on newstest14 and LibriSpeech.

| Model | NMT BLEU | | ASR % WER | | NMT NLL | | ASR NLL | |
|---|---|---|---|---|---|---|---|---|
| | EN-DE | EN-FR | LTC | LTO | EN-DE | EN-FR | LTC | LTO |
| Single | 28.8 ±0.2 | 45.4 ± 0.3 | 5.6 ±0.2 | 14.7 ±0.5 | 1.46 ± 0.02 | 1.10 ± 0.01 | 0.34 ±0.00 | 0.86 ±0.02 |
| ENS-PrEx | **30.1** | **46.5** | **4.2** | **11.3** | **1.33** | **1.04** | **0.20** | **0.48** |
| ENS-ExPr | 29.9 | 46.3 | 4.5 | 12.6 | 1.36 | 1.05 | 0.23 | 0.58 |

**Choice of Ensemble Combination** As discussed in section 3, ensembles can be combined as an *expectation-of-products* (ExPr) or as a *product-of-expectations* (PrEx) (16). Therefore, it is necessary to evaluate which yields superior predictive performance. We evaluate EN-DE and EN-FR NMT models on newstest14 and the ASR models on LibriSpeech test-clean (LTC) and test-other (LTO).

Results in table 1 show that a *product-of-expectations* combination consistently yields marginally higher translation BLEU and lower ASR word-error-rate (WER) in beam-search decoding for all tasks[4]. Beam-width for NMT and ASR models is 5 and 20, respectively. We speculate that this is because beam-search inference, which is a sequence of greedy *token-level* decisions, benefits more from token-level Bayesian model averaging. At the same time, both combination strategies yield equivalent teacher-forcing mean length-normalized negative-log-likelihood on reference data. This may be because the models in the ensemble yield consistent predictions on in-domain data, in which case the two combinations will yield similar probabilities. Further experiments in this work will use hypotheses obtained from a *product-of-expectations* ensemble combination, as it yields marginally better predictive performance. Additional analysis and results are available in appendix C.

Table 2: Sequence-level Error Detection % Prediction Rejection Ratio in Beam-Search decoding.

| Task | Test set | ENS-PrEx TU | | ENS-ExPr TU | | ENS-PrEx KU | | | | ENS-ExPr KU | | | |
|---|---|---|---|---|---|---|---|---|---|---|---|---|---|
| | | $\hat{\mathcal{H}}_{\text{C-IW}}^{(1)}$ | $\hat{\mathcal{H}}_{\text{S-IW}}^{(1)}$ | $\hat{\mathcal{H}}_{\text{C-IW}}^{(1)}$ | $\hat{\mathcal{H}}_{\text{S-IW}}^{(1)}$ | $\hat{\mathcal{I}}_{\text{C-IW}}^{(1)}$ | $\hat{\mathcal{K}}_{\text{C-IW}}^{(1)}$ | $\hat{\mathcal{M}}_{\text{C-IW}}^{(1)}$ | $\hat{\mathcal{M}}_{\text{S-IW}}^{(1)}$ | $\hat{\mathcal{I}}_{\text{C-IW}}^{(1)}$ | $\hat{\mathcal{K}}_{\text{C-IW}}^{(1)}$ | $\hat{\mathcal{M}}_{\text{C-IW}}^{(1)}$ | $\hat{\mathcal{M}}_{\text{S-IW}}^{(1)}$ |
| LSP | LTC | 61.2 | **65.9** | 60.5 | 64.9 | 59.0 | 57.2 | 56.5 | 61.0 | 55.1 | 56.5 | 56.2 | 60.2 |
| | LTO | 68.8 | **71.7** | 67.0 | 67.9 | 67.4 | 64.2 | 63.2 | 64.8 | 57.5 | 63.2 | 62.7 | 61.4 |
| | AMI | 57.2 | **66.8** | 52.3 | 61.5 | 54.2 | 51.8 | 50.6 | 63.5 | 25.9 | 49.2 | 49.0 | 56.4 |
| ENDE | nwt14 | 28.1 | **45.8** | 27.8 | 45.5 | 27.3 | 26.3 | 25.6 | 28.9 | 15.9 | 23.9 | 26.2 | 25.4 |
| ENFR | | 25.9 | **39.0** | 25.6 | 38.8 | 29.8 | 29.3 | 28.8 | 32.4 | 20.3 | 27.1 | 28.6 | 29.6 |

**Sequence-level Error Detection** We now investigate whether the sequence-level uncertainty measures can be used to detect sentences which are challenging to translate or transcribe. In the following experiment a model's 1-best hypotheses are sorted in order of decreasing uncertainty and incrementally replaced by the references. The mean *sentence-BLEU* (sBLEU) or *sentence-WER* (sWER) is plotted against the fraction of data replaced on a *rejection curve*. If the uncertainties are informative, then the increase in sBLEU or decrease in sWER should be greater than random (linear). Rejection curves are summarised using the *Prediction Rejection Ratio* (PRR) (Malinin, 2019; Malinin et al., 2020), describe in appendix D.2, which is 100% if uncertainty estimates perfectly correlate with sentence BLEU/WER, and 0% if they are uninformative. In these experiments information only from the 1-best hypothesis is considered.[5] While the 1-best hypotheses are obtained from a product-of-expectation combination, we consider uncertainty estimates obtained by expressing the predictive posterior both as a product-of-expectations (ENS-PrEx) and expectation-of-products (ENS-ExPr).

Table 2 shows several trends. First, measures of *total uncertainty* yield the best performance. Furthermore, joint-sequence estimates of *total uncertainty* consistently outperform chain-rule based estimates. This is because, unlike chain-rule approximations, joint-sequence approximations do not account for probabilities of non-generated tokens and only consider probabilities *along* the 1-best hypothesis, and therefore assess its quality. This is consistent with results for unstructured-prediction (Malinin, 2019). Second, measures derived from a *product-of-expectation* predictive

---

[4]BLEU was calculated using sacrebleu (Post, 2018) and WER using sclite.

[5]Assessment of uncertainty derived from all hypotheses in the beam are analyzed in appendix D.

posterior tend to yield superior performance than their *expectation-of-products* counterparts. However, this does not seem to be a property of the 1-best hypotheses, as results in appendix D on hypotheses obtained from a *expectation-of-products* ensemble show a similar trend. Third, out of all measures of knowledge uncertainty, joint-sequence RMI performs best. Finally, the performance gap between chain-rule and joint-sequence estimates of total uncertainty is larger for NMT. This is because compared to ASR, NMT is a task with intrinsically higher uncertainty, and therefore more irrelevant information is introduced by considering the probabilities of non-generated tokens.

The results also show that uncertainty-based rejection works better for ASR than NMT. The issue lies in the nature of NMT - it is inherently difficult to objectively define a bad translation[6]. While WER is an objective measure of quality, BLEU is only a proxy measure. While a high sBLEU indicates a good translation, a low sBLEU does not necessarily indicate a poor one. Thus, a model may yield a low uncertainty, high-quality translation which has little word-overlap with the reference and low sBLEU, negatively impacting PRR. A better, but more expensive, approach to assess uncertainty estimates in NMT is whether they correlate well with human assessment of translation quality.

Table 3: Token-level Error Detection %AUPR for LibriSpeech in Beam-Search Decoding regime.

| Test Data | ENSM-PrEx TU $\mathcal{H}$ | ENSM-PrEx TU $\mathcal{P}$ | ENSM-ExPr TU $\mathcal{H}$ | ENSM-ExPr TU $\mathcal{P}$ | ENS-PrEx KU $\mathcal{I}$ | ENS-PrEx KU $\mathcal{K}$ | ENS-PrEx KU $\mathcal{M}$ | ENS-ExPr KU $\mathcal{I}$ | ENS-ExPr KU $\mathcal{K}$ | ENS-ExPr KU $\mathcal{M}$ | % TER |
|---|---|---|---|---|---|---|---|---|---|---|---|
| LTC | 34.7 | **36.3** | 32.4 | 33.4 | 32.7 | 28.0 | 27.6 | 26.5 | 25.9 | 27.4 | 3.8 |
| LTO | 42.4 | **43.3** | 39.0 | 39.1 | 40.9 | 37.1 | 36.1 | 30.8 | 33.6 | 35.3 | 10.2 |
| AMI | 71.7 | **74.6** | 68.3 | 70.4 | 71.8 | 67.9 | 68.7 | 59.2 | 64.4 | 66.6 | 41.2 |

**Token-level Error Detection** We now assess whether token-level uncertainties can be used to detect token-level errors in the models' 1-best hypotheses. Note that token-level error labelling is ill-posed for translation, where correct tokens can be mislabelled as errors due valid word re-arrangements and substitutions. Thus, token-level error detection is only investigated for ASR. Ground-truth error-labels are obtained by aligning the hypotheses to the references using the SCLITE NIST scoring tool and marking insertions and substitutions[7]. Performance is assessed via area-under a Precision-Recall curve. Random performance corresponds to the baseline recall, which is equal to the token error rate. Results in table 3 are consistent with the previous section. First, measures of *total uncertainty* outperform measures of *knowledge uncertainty*. Second, estimates derived from conditional log-scores $\mathcal{P}$ of the generated token outperform the entropy $\mathcal{H}$ of the token-level predictive posterior. This is because the latter relates to probability of an error *at this position*, while the former relates to the probability of *this particular token* being an error. Finally, deriving uncertainties from a product-of-expectation token-level predictive posterior $\mathrm{P}_{\mathrm{PE}}(y_l|\boldsymbol{y}^{(1)}_{<l}, \boldsymbol{x}, \mathcal{D})$ yields superior results.

Table 4: OOD Detection % ROC-AUC in Beam-Search decoding regime for ASR and NMT.

| Task | OOD Data | $T$ | $B$ | ENS-PrEx TU $\hat{\mathcal{H}}^{(B)}_{C\text{-}IW}$ | ENS-PrEx TU $\hat{\mathcal{H}}^{(B)}_{S\text{-}IW}$ | ENS-ExPr TU $\hat{\mathcal{H}}^{(B)}_{C\text{-}IW}$ | ENS-ExPr TU $\hat{\mathcal{H}}^{(B)}_{S\text{-}IW}$ | ENS-PrEx KU $\hat{\mathcal{I}}^{(B)}_{C\text{-}IW}$ | ENS-PrEx KU $\hat{\mathcal{K}}^{(B)}_{C\text{-}IW}$ | ENS-PrEx KU $\hat{\mathcal{M}}^{(B)}_{C\text{-}IW}$ | ENS-PrEx KU $\hat{\mathcal{M}}^{(B)}_{S\text{-}IW}$ | ENS-ExPr KU $\hat{\mathcal{I}}^{(B)}_{C\text{-}IW}$ | ENS-ExPr KU $\hat{\mathcal{K}}^{(B)}_{C\text{-}IW}$ | ENS-ExPr KU $\hat{\mathcal{M}}^{(B)}_{C\text{-}IW}$ | ENS-ExPr KU $\hat{\mathcal{M}}^{(B)}_{S\text{-}IW}$ |
|---|---|---|---|---|---|---|---|---|---|---|---|---|---|---|---|
| ASR | LTO | 1 | 1 | 76.7 | 75.5 | 76.2 | 75.0 | 76.4 | 76.6 | 76.6 | 73.9 | 74.0 | 76.3 | 76.4 | 73.4 |
|  |  |  | 20 | 76.9 | 76.3 | 76.4 | **77.0** | 76.6 | **77.0** | **77.0** | 76.1 | 74.8 | 76.7 | 76.9 | 75.3 |
|  | AMI | 1 | 1 | 97.5 | 97.6 | 97.0 | 97.2 | 96.4 | 96.2 | 96.2 | 96.4 | 90.1 | 95.7 | 95.9 | 95.8 |
|  |  |  | 20 | 96.5 | **97.9** | 96.4 | **97.9** | 94.9 | 94.9 | 94.8 | 97.4 | 93.0 | 94.8 | 94.8 | 97.0 |
|  | C-FR | 1 | 1 | **99.9** | 99.7 | 99.8 | 99.7 | **99.9** | **99.9** | **99.9** | 99.8 | 81.0 | 99.8 | 99.8 | 98.8 |
|  |  |  | 20 | **99.9** | 99.7 | **99.9** | 99.8 | **99.9** | **99.9** | **99.9** | 99.8 | 89.7 | **99.9** | **99.9** | 99.1 |
| NMT | LTC | 10 | 1 | 65.7 | 71.8 | 64.7 | 71.0 | 72.8 | 72.5 | 72.2 | 73.3 | 57.1 | 68.4 | 71.8 | 71.9 |
|  |  |  | 5 | 66.0 | 74.1 | 65.0 | 74.0 | 73.2 | 72.9 | 72.6 | **75.0** | 58.1 | 69.3 | 72.1 | 73.9 |
|  | PRM | 10 | 1 | 82.2 | 82.7 | 79.8 | 83.5 | 96.4 | 96.6 | 96.7 | 96.2 | 69.3 | 93.9 | 96.4 | 94.2 |
|  |  |  | 5 | 82.9 | 82.7 | 80.7 | 84.5 | 96.7 | **96.9** | **96.9** | 96.5 | 72.1 | 95.0 | 96.7 | 95.2 |
|  | L-FR | 10 | 1 | 26.4 | 18.5 | 22.2 | 18.6 | 63.0 | 68.7 | 72.1 | 70.9 | 22.7 | 44.9 | 69.9 | 76.4 |
|  |  |  | 5 | 27.1 | 21.7 | 23.0 | 22.9 | 65.2 | 71.2 | 74.8 | **79.6** | 23.4 | 50.1 | 73.8 | 80.7 |
|  | L-DE | 10 | 1 | 39.8 | 28.7 | 35.1 | 30.2 | 74.4 | 78.0 | 80.1 | 76.0 | 41.1 | 68.8 | 78.4 | 77.4 |
|  |  |  | 5 | 40.8 | 34.9 | 36.1 | 38.3 | 76.1 | 79.9 | 82.1 | **89.2** | 41.9 | 73.3 | 81.2 | 88.8 |

**Out-of-Domain input Detection** We now consider out-of-domain input (anomaly) detection. The goal is use uncertainty estimates to discriminate between in-domain test data and a selection of out-

---

[6]Provided that the model is high-performing in general.

[7]Detecting deletions is, in general, a far more challenging task.

of-domain (OOD) datasets. Performance is assessed via area under a ROC-curve (ROC-AUC), where 100% is ideal performance, 50% is random and below 50% indicates that the model yields lower uncertainty for the OOD data. Results are presented in table 4, additional results in appendices F,G,I.

First, let's examine OOD detection for speech recognition. Three OOD datasets are considered, each covering a different form of domain shift. First, LibriSpeech test-other (LTO), which represents a set of sentences which are more noisy and difficult to transcribe. Second, the AMI meeting transcription dataset (Kraaij et al., 2005), which represents spoken English from a different domain, mismatched to LibriSpeech, which consist of books being read. Finally, we consider speech in a different language (French), taken from the Common Voice Project (Ardila et al., 2019). The results show that OOD detection becomes easier the greater the domain mismatch. Curiously, there is marginal difference between the performance of measures of uncertainty. This is likely because ASR models tend to be very 'sharp' and are naturally more entropic in mismatched conditions.

Now let's consider OOD detection for WMT'17 English-German machine translation. The following OOD datasets are considered. First, the LibriSpeech test-clean (LTC) reference transcriptions, which are OOD in terms of both domain and structure, as spoken English is structurally distinct from written English. Second, newstest14 sentences corrupted by randomly permuting the source-tokens (PRM). Third, French and German source-sentences from newstest14 (L-FR and L-DE). Results show that discriminating between spoken and written English is challenging. In contrast, it is possible to near-perfectly detect corrupted English. Interestingly, detection of text from other languages is particularly difficult. Inspection of the model's output shows that the ensemble displays a pathological copy-through effect, where the input tokens are copied to the output with high confidence. As a result, estimates of *total uncertainty* are lower for the (OOD) French or German data than for (ID) English data. Notably, estimates of *knowledge uncertainty*, especially reverse mutual information (RMI), $\hat{\mathcal{M}}_{\text{C-IW}}^{(B)}$ and $\hat{\mathcal{M}}_{\text{S-IW}}^{(B)}$, are affected far less and discriminate between the in-domain and OOD data. This effect true in general, but is especially pronounced when the copy-through effect is triggered. This highlights the value of the RMI, for which asymptotically exact approximations can be obtained.

Clearly, ASR ensembles are better at OOD detection than NMT ensembles. This is expected, as ASR models receive a continuous-valued input signal which contains information not only about the content of the speech, but also the domain, language, speaker characteristics, background noise and recording conditions. This makes the task easier, as the model conditions on more information. This is also why ASR has low intrinsic *data uncertainty* and why the best OOD detection performance for ASR is obtained using measures of *total uncertainty*. In contrast, NMT models only have access to a sequence of discrete tokens, which contains far less information. This also highlights the value of *knowledge uncertainty*, as it disregard the high intrinsic *data uncertainty* of NMT.

An interesting effect, which is fully explored in appendix G, is that when considering all the hypotheses within the beam for uncertainty estimation, it is beneficial for NMT to use a higher importance-weighting temperature ($T = 10$), increasing the contribution from competing hypotheses. In contrast, this *detrimental* for ASR, and temperature is kept at $T = 1$. We hypothesise that this may be an artifact of the *multi-modality* of translation - multiple hypotheses could be equally good and contribute valuable information. In contrast, in ASR there is only one correct transcription, though not necessarily the 1-best, and considering competing hypotheses is detrimental.

The results also show that chain-rule and joint-sequence approximations yield similar performance and that, with the exception of $\hat{\mathcal{M}}_{\text{S-IW}}^{(B)}$, using information from the full beam yields benefits minor improvements compared during using just the 1-best hypotheses. Uncertainties derived from $P_{\text{EP}}(\boldsymbol{y}|\boldsymbol{x}, \mathcal{D})$ and $P_{\text{PE}}(\boldsymbol{y}|\boldsymbol{x}, \mathcal{D})$ yield comparable performance, with the exception of mutual information and EPKL, where $P_{\text{EP}}(\boldsymbol{y}|\boldsymbol{x}, \mathcal{D})$ yields consistently poorer performance. This suggests that $P_{\text{PE}}(\boldsymbol{y}|\boldsymbol{x}, \mathcal{D})$ yields more robust uncertainty estimates.

**Comparison to heuristic uncertainty measures** We close with a comparison of the proposed information-theoretic measures of *knowledge uncertainty* to the 'heuristic' measures describes in (Wang et al., 2019; Fomicheva et al., 2020; Xiao et al., 2019). These measures, and our modifications thereof, are detailed in appendix I. We examine the variance of the length-normalized probability and log-probability of hypotheses across the ensemble, as well as the cross-hypothesis WER/BLEU. The use of more than the 1-best hypothesis is our extension for the variance-based measures.

All of these measures aim to evaluate the diversity of ensemble of models in different ways. In this regard they are all measures of *knowledge uncertainty*. Their main limitations, as originally presented, are the following. All measures focused on the diversity in the probabilities or surface forms of only the 1-best hypothesis. While sufficient in some tasks, such as sequence-error detection, this prevents them from fully capture information about the behavior of the space of possible translations/transcriptions. In this regard, the information theoretic measures presented in our work are an advantage, as they naturally allow to do this. We attempt to address this for the variance-based measures by considering the importance-weighted variance of the probability/log-probability of each hypothesis in the beam. While not strictly rigorous, this nonetheless attempts to address the problem. Such an extension to cross-BLEU/WER is not possible, as it is not clear how to match up different hypotheses across all decodings of each model in the ensemble. Cross-BLEU/WER have the additional limitation of needing a separate decoding of *each model* in the ensemble, which is undesirable and expensive. Finally, it is likely that there is bias towards longer hypotheses as being more diverse, as there is a greater chance of a surface form mismatch.

The results in table 5 show that information-theoretic measures consistently yield better performance, though sometimes only marginally. Cross-BLEU/WER typically yields the worst performance, especially for NMT. Finally, including information from competing hypotheses can be advantageous for the variance based measures. However, sometimes the performance is degraded - this is because the information was integrated in an adhoc, rather than theoretically meaningful, fashion. We also show in appendix I that length-normalization, which was used inconsistently in prior work, is important both for these measures, and appendix H for the information-theoretic ones.

Table 5: Comparison of info-theoretic and heuristic measures on OOD detection (% ROC-AUC).

| Task | OOD Data | $T$ | $B$ | Info.Theor. | | Heuristic | | | |
|------|----------|-----|-----|-------------|--------------|------------------|------------------------|--------|--------|
| | | | | $\hat{\mathcal{M}}_{\text{C-IW}}^{(B)}$ | $\hat{\mathcal{M}}_{\text{S-IW}}^{(B)}$ | $\hat{\mathbb{V}}^{(B)}[P]$ | $\hat{\mathbb{V}}^{(B)}[\ln P]$ | X-BLEU | X-WER |
| ASR | LTO | 1 | 1 | 76.6 | 73.9 | 72.0 | 72.7 | 74.3 | 71.8 |
| | | | 20 | **77.0** | 76.1 | 72.7 | 74.6 | | |
| | AMI | 1 | 1 | 96.2 | 96.4 | 87.3 | 95.8 | 95.9 | 95.8 |
| | | | 20 | 94.8 | **97.4** | 85.6 | 96.7 | | |
| | C-FR | 1 | 1 | **99.9** | 99.8 | 82.0 | 98.0 | 99.5 | 99.7 |
| | | | 20 | **99.9** | 99.8 | 77.6 | 98.4 | | |
| NMT | LTC | 10 | 1 | 72.2 | 73.3 | 68.7 | 72.3 | 65.1 | 58.2 |
| | | | 5 | 72.6 | **75.0** | 68.6 | 72.5 | | |
| | PRM | 10 | 1 | 96.7 | 96.2 | 88.8 | 93.4 | 80.8 | 73.5 |
| | | | 5 | **96.9** | 96.5 | 88.7 | 93.0 | | |
| | L-FR | 10 | 1 | 72.1 | 70.9 | 76.4 | 71.1 | 46.8 | 52.7 |
| | | | 5 | 74.8 | **79.6** | 77.7 | 72.2 | | |

## 5 CONCLUSION

This work investigated applying a general, probabilistically interpretable ensemble-based uncertainty estimation framework to structured tasks, focusing on autoregressive models. A range of information-theoretic uncertainty measures both at the *token level* and *sequence level* were considered, including a novel measure of knowledge uncertainty called reverse mutual-information (RMI). Two types of Monte-Carlo approximations were proposed - one based on the entropy chain rule, and the other on sequence samples. Additionally, this work examined ensemble combination through both token-level and sequence-level Bayesian model averaging. Performance baselines for sequence and token-level error detection, and out-of-domain (OOD) input detection were provided on the WMT'14 English-French and WMT'17 English-German translation datasets, and the LibriSpeech ASR dataset. The results show that ensemble-based measures of uncertainty are useful for all applications considered. Estimates of *knowledge uncertainty* are especially valuable for NMT OOD detection. Crucially, it was shown that RMI is consistently the most informative measure of knowledge uncertainty for structured prediction. Notably, it was found that token-level Bayesian model averaging consistently yields both marginally better predictive performance and more robust estimates of uncertainty. However, it remains unclear why this is the case, which should be investigated in future work. Future work should also investigate alternative ensemble generation techniques and compare ensemble-based uncertainty estimates to the task-specific confidence-score estimates previously explored for ASR and NMT. Another interesting direction is to assess the calibration of autoregressive ASR models.

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

# A DERIVATIONS OF UNCERTAINTY MEASURES

The current appendix details token-level measures of uncertainty for autoregressive models and provides the derivations of the sequence-level measures of uncertainty discussed in section 3 as well as extended discussions of their theoretical properties.

## A.1 TOKEN-LEVEL UNCERTAINTY ESTIMATES

As was stated in section 2, token-level ensemble-based uncertainty estimates for autoregressive models are isomorphic to un-structured uncertainty estimates (Malinin, 2019). However, for completeness, they are described in the current section.

First, let's consider the predictive posterior $P(y_l|\boldsymbol{y}_{<l}, \boldsymbol{x}, \mathcal{D})$. As discussed in section 3, the token-level predictive posterior of an autoregressive models can be obtained in two ways. One corresponds to *token-level* Bayesian model averaging, while the other corresponds to *sequence-level* Bayesian model averaging. The first can be expressed as follows:

$$P(y_l|\boldsymbol{y}_{<l}, \boldsymbol{x}, \mathcal{D}) = \mathbb{E}_{\mathsf{q}(\boldsymbol{\theta})}\big[P(y_l|\boldsymbol{y}_{<l}, \boldsymbol{x}, \boldsymbol{\theta})\big] = \frac{1}{M}\sum_{m=1}^{M} P(y_l|\boldsymbol{y}_{<l}, \boldsymbol{x}, \boldsymbol{\theta}^{(m)}), \ \boldsymbol{\theta}^{(m)} \sim \mathsf{q}(\boldsymbol{\theta}) \quad (18)$$

While the latter is expressed like so:

$$P(y_l|\boldsymbol{y}_{<l}, \boldsymbol{x}, \mathcal{D}) = \frac{P(y_l, \boldsymbol{y}_{<l}, \boldsymbol{x}, \mathcal{D})}{P(\boldsymbol{y}_{<l}, \boldsymbol{x}, \mathcal{D}} = \frac{\mathbb{E}_{\mathsf{q}(\boldsymbol{\theta})}[P(y_l, \boldsymbol{y}_{<l}, \boldsymbol{x}, \boldsymbol{\theta})]}{\mathbb{E}_{\mathsf{q}(\boldsymbol{\theta})}[P(\boldsymbol{y}_{<l}, \boldsymbol{x}, \boldsymbol{\theta}]} \approx \frac{\frac{1}{M}\sum_m P(y_l, \boldsymbol{y}_{<l}, \boldsymbol{x}, \boldsymbol{\theta}^{(m)})}{\frac{1}{M}\sum_m P(\boldsymbol{y}_{<l}, \boldsymbol{x}, \boldsymbol{\theta}^{(m)})} \quad (19)$$

Clearly, token-level BMA is more consistent with estimating the uncertainty in the prediction of the current token $y_l$, regardless of how the context tokens were generated. In contrast, sequence-level BMA considers how the entire sequence was generated.

Regardless of how the predictive posterior is obtained, the estimate of *total uncertainty* will be given its entropy:

$$\underbrace{\mathcal{H}\big[P(y_l|\boldsymbol{y}_{<l}, \boldsymbol{x}, \mathcal{D})\big]}_{\text{Total Uncertainty}} = -\sum_{k=1}^{K} P(y_l = \omega_k|\boldsymbol{y}_{<l}, \boldsymbol{x}, \mathcal{D}) \ln P(y_l = \omega_k|\boldsymbol{y}_{<l}, \boldsymbol{x}, \mathcal{D}) \quad (20)$$

Furthermore, by considering the mutual information between $y_l$ and $\boldsymbol{\theta}$ we can obtain measures of *total uncertainty*, *knowledge uncertainty* and *expected data uncertainty*:

$$\underbrace{\mathcal{I}\big[y_l, \boldsymbol{\theta}|\boldsymbol{y}_{<l}, \boldsymbol{x}, \mathcal{D}\big]}_{\text{Knowledge Uncertainty}} = \mathbb{E}_{\mathsf{q}(\boldsymbol{\theta})}\Big[\mathbb{E}_{P(y_l|\boldsymbol{y}_{<l}, \boldsymbol{x}; \boldsymbol{\theta})}\Big[\ln \frac{P(y_l|\boldsymbol{y}_{<l}, \boldsymbol{x}; \boldsymbol{\theta})}{P(y_l|\boldsymbol{y}_{<l}, \boldsymbol{x}; \mathcal{D})}\Big]\Big]$$

$$= \underbrace{\mathcal{H}\big[P(y_l|\boldsymbol{y}_{<l}, \boldsymbol{x}, \mathcal{D})\big]}_{\text{Total Uncertainty}} - \underbrace{\mathbb{E}_{\mathsf{q}(\boldsymbol{\theta})}\big[\mathcal{H}[P(y_l|\boldsymbol{y}_{<l}, \boldsymbol{x}; \boldsymbol{\theta})]\big]}_{\text{Expected Data Uncertainty}} \quad (21)$$

Alternatively, the expected pair-wise KL-divergence (EPKL) between models in the ensemble at the token level can also be considered:

$$\underbrace{\mathcal{K}[y_l, \boldsymbol{\theta}|\boldsymbol{y}_{<l}, \boldsymbol{x}, \mathcal{D}]}_{\text{Knowledge Uncertainty}} = \mathbb{E}_{\mathsf{q}(\boldsymbol{\theta})\mathsf{q}(\tilde{\boldsymbol{\theta}})}\big[\text{KL}[P(y_l|\boldsymbol{y}_{<l}, \boldsymbol{x}, \boldsymbol{\theta})||P(y_l|\boldsymbol{y}_{<l}, \boldsymbol{x}, \tilde{\boldsymbol{\theta}})]\big]$$

$$= \mathbb{E}_{P(y_l|\boldsymbol{y}_{<l}, \boldsymbol{x}, \mathcal{D})}\big[\mathbb{E}_{\mathsf{q}(\tilde{\boldsymbol{\theta}})}[-\ln P(y_l|\boldsymbol{y}_{<l}, \boldsymbol{x}; \tilde{\boldsymbol{\theta}})]\big] - \underbrace{\mathbb{E}_{\mathsf{q}(\boldsymbol{\theta})}\big[\mathcal{H}[P(y_l|\boldsymbol{y}_{<l}, \boldsymbol{x}; \boldsymbol{\theta})]\big]}_{\text{Expected Data Uncertainty}} \quad (22)$$

where $\mathsf{q}(\boldsymbol{\theta}) = \mathsf{q}(\tilde{\boldsymbol{\theta}})$. This yields an alternative measure of ensemble diversity which is a Jensen-derived upper bound on mutual information. Both EPKL and mutual information yield the same estimate of *data uncertainty*. We can also consider novel measures of diversity, and therefore *knowledge uncertainty*, called reverse mutual information (RMI) $\mathcal{M}$ defined as follows:

$$\underbrace{\mathcal{M}\big[y_l, \boldsymbol{\theta}|\boldsymbol{y}_{<l}, \boldsymbol{x}, \mathcal{D}\big]}_{\text{Knowledge Uncertainty}} = \mathbb{E}_{\mathsf{q}(\boldsymbol{\theta})}\Big[\mathbb{E}_{P(y_l|\boldsymbol{y}_{<l}, \boldsymbol{x}; \mathcal{D})}\Big[\ln \frac{P(y_l|\boldsymbol{y}_{<l}, \boldsymbol{x}; \mathcal{D})}{P(y_l|\boldsymbol{y}_{<l}, \boldsymbol{x}; \boldsymbol{\theta})}\Big]\Big]$$

$$= \mathbb{E}_{P(y_l|\boldsymbol{y}_{<l}, \boldsymbol{x}, \mathcal{D})}\big[\mathbb{E}_{\mathsf{q}(\boldsymbol{\theta})}[-\ln P(y_l|\boldsymbol{y}_{<l}, \boldsymbol{x}; \boldsymbol{\theta})]\big] - \underbrace{\mathcal{H}\big[P(y_l|\boldsymbol{y}_{<l}, \boldsymbol{x}, \mathcal{D})\big]}_{\text{Total Uncertainty}} \quad (23)$$

This is effectively the reverse-KL divergence counterpart to the mutual information, which is the mean KL-divergence between each model and the predictive posterior. It yields the same estimates of *total uncertainty*. Just like for the sequence-level measures of uncertainty, it is trivial to derive a relationship between these token-level measures of ensemble diversity:

$$\mathcal{M}\big[y_l, \boldsymbol{\theta}|\boldsymbol{y}_{<l}, \boldsymbol{x}, \mathcal{D}\big] = \mathcal{K}\big[y_l, \boldsymbol{\theta}|\boldsymbol{y}_{<l}, \boldsymbol{x}, \mathcal{D}\big] - \mathcal{I}\big[y_l, \boldsymbol{\theta}|\boldsymbol{y}_{<l}, \boldsymbol{x}, \mathcal{D}\big] \tag{24}$$

Thus, RMI is the difference between the EPKL and mutual information. Consequently, while mutual information, EPKL and RMI all yield estimates of *knowledge uncertainty*, only mutual information 'cleanly' decomposes into *total uncertainty* and *data uncertainty*. In contrast, EPKL *does not* yield clean measures of *total uncertainty* and RMI does not yield clean measures of *data uncertainty*.

All of these measures of uncertainty considered above use information from the full distribution over tokens $y_l$. However, we can also examine measures which only consider the probability assigned to the predicted token $\hat{\omega}_l$. Firstly, we can examine the log-likelihood of the predicted token under the predictive posterior:

$$\mathcal{P} = -\ln \mathrm{P}(y_l = \hat{\omega}_l|\boldsymbol{y}_{<l}, \boldsymbol{x}, \mathcal{D}) \tag{25}$$

This is a measure of *total uncertainty*. Alternatively, we can consider the mean negative *Point-wise Mutual Information* (Murphy, 2012) between the model $\boldsymbol{\theta}^{(m)}$ and the prediction $\boldsymbol{y}_l$ across all models:

$$\mathcal{M}_{\omega_l} = -\mathbb{E}_{\mathsf{q}(\boldsymbol{\theta})}\underbrace{\left[\ln \frac{\mathrm{P}(y_l = \hat{\omega}_l|\boldsymbol{y}_{<l}, \boldsymbol{x}, \boldsymbol{\theta})}{\mathrm{P}(y_l = \hat{\omega}_l|\boldsymbol{y}_{<l}, \boldsymbol{x}, \mathcal{D})}\right]}_{\text{Pointwise Mutual Information}} \tag{26}$$

This is effectively the RMI at the point prediction, rather than an expectation over all classes.

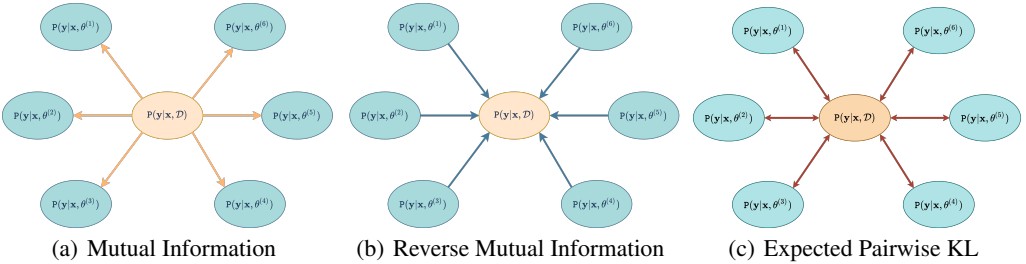

(a) Mutual Information    (b) Reverse Mutual Information    (c) Expected Pairwise KL

Figure 1: Measures of Ensemble Diversity - Direction of KL Divergence

For autoregressive models, these measures represent uncertainty in the prediction given a specific combination of input $\boldsymbol{x}$ and context $\boldsymbol{y}_{<l}$. However, at the beginning of a sequence token-level measures of uncertainty are more sensitive to the input $\boldsymbol{x}$ and at the end of a sequence become more sensitive to the context $\boldsymbol{y}_{<l}$.

## A.2 Derivation of Sequence-level Monte-Carlo approximations

In the current section we detail the derivations of joint-sequence and chain-rule derived Monte-Carlo approximations of sequence-level measures of uncertainty defined in 3. Crucially, they make use of the chain rules of entropy and relative entropy (Cover & Thomas, 2006):

$$\hat{\mathcal{H}}[\mathrm{P}(\boldsymbol{y}|\boldsymbol{x}, \boldsymbol{\theta})] = \frac{1}{L}\sum_{l=1}^{L}\mathbb{E}_{\mathrm{P}(\boldsymbol{y}_{<l}|\boldsymbol{x}, \boldsymbol{\theta})}\big[\mathcal{H}[\mathrm{P}(y_l|\boldsymbol{x}, \boldsymbol{y}_{<l}, \boldsymbol{\theta})]\big] \tag{27}$$

$$\hat{\mathrm{KL}}[\mathrm{P}(\boldsymbol{y}|\boldsymbol{x})\|\mathrm{Q}(\boldsymbol{y}|\boldsymbol{x})] = \frac{1}{L}\sum_{l=1}^{L}\mathbb{E}_{\mathrm{P}(\boldsymbol{y}_{<l}|\boldsymbol{x})}\big[\mathrm{KL}[\mathrm{P}(y_l|\boldsymbol{y}_{<l}, \boldsymbol{x})\|\mathrm{Q}(y_l|\boldsymbol{y}_{<l}, \boldsymbol{x})]\big] \tag{28}$$

We omit the derivation for the entropy of the predictive posterior as they are straightforward. Instead, we focus on measures of *knowledge uncertainty*. First, lets consider a direct joint-sequence Monte-Carlo estimate for mutual information:

$$\hat{\mathcal{I}}^{(S)}\big[\boldsymbol{y}, \boldsymbol{\theta}|\boldsymbol{x}, \mathcal{D}\big] \approx \frac{1}{S}\sum_{s=1}^{S}\frac{1}{L^{(s)}}\ln\frac{\mathrm{P}(\boldsymbol{y}^{(s)}|\boldsymbol{x}, \boldsymbol{\theta}^{(s)})}{\mathrm{P}(\boldsymbol{y}^{(s)}|\boldsymbol{x}, \mathcal{D})}, \ \boldsymbol{y}^{(s)} \sim \mathrm{P}(\boldsymbol{y}|\boldsymbol{x}, \boldsymbol{\theta}^{(s)}), \boldsymbol{\theta}^{(s)} \sim \mathsf{q}(\boldsymbol{\theta}) \tag{29}$$

Clearly, this is *inference inefficient*, as it requires independently sampling from *each* model. As a result, we cannot obtain this estimate of mutual information *for free* during sampling (or Beam-Search decoding) from the ensemble's predictive posterior. However, it is possible to obtain an inference efficient approximation by consider the chain rule of relative entropy:

$$\hat{\mathcal{I}}\big[\boldsymbol{y},\boldsymbol{\theta}|\boldsymbol{x},\mathcal{D}\big] = \mathbb{E}_{\mathsf{q}(\boldsymbol{\theta})}\Big[\frac{1}{L}\sum_{l=1}^{L}\mathbb{E}_{\mathsf{P}(\boldsymbol{y}_{<l}|\boldsymbol{x},\boldsymbol{\theta})}\big[\mathrm{KL}[\mathsf{P}(y_l|\boldsymbol{y}_{<l},\boldsymbol{x},\boldsymbol{\theta})\|\mathsf{P}(y_l|\boldsymbol{y}_{<l},\boldsymbol{x},\mathcal{D})]\big]\Big] \qquad (30)$$

Here, we have expressed mutual information as a sum of expected token-level KL-divergences. However, by replacing the expectation with respect to each individual model by the expectation with respect to the predictive posterior, we obtain the following approximation:

$$\begin{aligned}\hat{\mathcal{I}}\big[\boldsymbol{y},\boldsymbol{\theta}|\boldsymbol{x},\mathcal{D}\big] &\approx \mathbb{E}_{\mathsf{q}(\boldsymbol{\theta})}\Big[\frac{1}{L}\sum_{l=1}^{L}\mathbb{E}_{\mathsf{P}(\boldsymbol{y}_{<l}|\boldsymbol{x},\mathcal{D})}\big[\mathrm{KL}[\mathsf{P}(y_l|\boldsymbol{y}_{<l},\boldsymbol{x},\boldsymbol{\theta})\|\mathsf{P}(y_l|\boldsymbol{y}_{<l},\boldsymbol{x},\mathcal{D})]\big]\Big] \\ &= \frac{1}{L}\sum_{l=1}^{L}\mathbb{E}_{\mathsf{P}(\boldsymbol{y}_{<l}|\boldsymbol{x},\mathcal{D})}\big[\mathcal{I}[y_l,\boldsymbol{\theta}|\boldsymbol{y}_{<l},\boldsymbol{x},\mathcal{D}]\big] \qquad (31) \\ &\approx \frac{1}{S}\sum_{s=1}^{S}\frac{1}{L^{(s)}}\sum_{l=1}^{L^{(s)}}\mathcal{I}[y_l,\boldsymbol{\theta}|\boldsymbol{y}_{<l}^{(s)},\boldsymbol{x},\mathcal{D}], \quad \forall \boldsymbol{y}_{<l}^{(s)} \subset \boldsymbol{y}^{(s)} \sim \mathsf{P}(\boldsymbol{y}|\boldsymbol{x},\mathcal{D})\end{aligned}$$

This reduces to the sum of *token-level* mutual information along hypotheses drawn from the ensemble's predictive posterior. However, this approximation will not longer yield equation 3 in the limit as $S \to \infty$. Nevertheless, this approximation, while inexact, may still be useful in practice. We can also examine sequence-level EPKL. Similar to the exact Monte-Carlo estimate of mutual information, exact Monte-Carlo estimation for EPKL is also inference in-efficient, as it requires sampling for all models individually:

$$\hat{\mathcal{K}}^{(S)}\big[\boldsymbol{y},\boldsymbol{\theta}|\boldsymbol{x},\mathcal{D}\big] \approx \frac{1}{S}\sum_{s=1}^{S}\frac{1}{L^{(s)}}\ln\frac{\mathsf{P}(\boldsymbol{y}^{(s)}|\boldsymbol{x},\boldsymbol{\theta}^{(s)})}{\mathsf{P}(\boldsymbol{y}^{(s)}|\boldsymbol{x},\tilde{\boldsymbol{\theta}}^{(s)})}, \; \boldsymbol{y}^{(s)} \sim \mathsf{P}(\boldsymbol{y}|\boldsymbol{x},\boldsymbol{\theta}^{(s)}), \boldsymbol{\theta}^{(s)}{\sim}\mathsf{q}(\boldsymbol{\theta}), \tilde{\boldsymbol{\theta}}^{(s)}{\sim}\mathsf{q}(\tilde{\boldsymbol{\theta}})$$

$$(32)$$

As before, we can use the chain-rule of relative entropy and replace sampling from each individual model with sampling from the predictive posterior:

$$\begin{aligned}\hat{\mathcal{K}}^{(S)}\big[\boldsymbol{y},\boldsymbol{\theta}|\boldsymbol{x},\mathcal{D}\big] &= \mathbb{E}_{\mathsf{q}(\boldsymbol{\theta})\mathsf{q}(\tilde{\boldsymbol{\theta}})}\Big[\frac{1}{L}\sum_{l=1}^{L}\mathbb{E}_{\mathsf{P}(\boldsymbol{y}_{<l}|\boldsymbol{x},\boldsymbol{\theta})}\big[\mathrm{KL}[\mathsf{P}(y_l|\boldsymbol{y}_{<l},\boldsymbol{x},\boldsymbol{\theta})\|\mathsf{P}(y_l|\boldsymbol{y}_{<l},\boldsymbol{x},\tilde{\boldsymbol{\theta}})]\big]\Big] \\ &\approx \mathbb{E}_{\mathsf{q}(\boldsymbol{\theta})\mathsf{q}(\tilde{\boldsymbol{\theta}})}\Big[\frac{1}{L}\sum_{l=1}^{L}\mathbb{E}_{\mathsf{P}(\boldsymbol{y}_{<l}|\boldsymbol{x},\mathcal{D})}\big[\mathrm{KL}[\mathsf{P}(y_l|\boldsymbol{y}_{<l},\boldsymbol{x},\boldsymbol{\theta})\|\mathsf{P}(y_l|\boldsymbol{y}_{<l},\boldsymbol{x},\tilde{\boldsymbol{\theta}})]\big]\Big] \\ &= \frac{1}{L}\sum_{l=1}^{L}\mathbb{E}_{\mathsf{P}(\boldsymbol{y}_{<l}|\boldsymbol{x},\mathcal{D})}\big[\mathcal{K}[y_l,\boldsymbol{\theta}|\boldsymbol{y}_{<l},\boldsymbol{x},\mathcal{D}]\big] \qquad (33) \\ &\approx \frac{1}{S}\sum_{s=1}^{S}\frac{1}{L^{(s)}}\sum_{l=1}^{L^{(s)}}\mathcal{K}[y_l,\boldsymbol{\theta}|\boldsymbol{y}_{<l}^{(s)},\boldsymbol{x},\mathcal{D}], \quad \forall \boldsymbol{y}_{<l}^{(s)} \subset \boldsymbol{y}^{(s)} \sim \mathsf{P}(\boldsymbol{y}|\boldsymbol{x},\mathcal{D})\end{aligned}$$

This approximation becomes the sum of token-level EPKL along hypotheses generated by the ensemble's predictive posterior. This approximation will also not yield equation 4 in the limit as $S \to \infty$.

However, while asymptotically exact inference-efficient Monte-Carlo estimates of sequence-level *knowledge uncertainty* cannot be obtained via mutual information and EPKL, they can by considering the novel measure RMI equation 5, which defined as an expectation with respect to the predictive posterior. A direct joint-sequence Monte-Carlo approximation can be obtained as follows:

$$\hat{\mathcal{M}}^{(S)}\big[\boldsymbol{y},\boldsymbol{\theta}|\boldsymbol{x},\mathcal{D}\big] \approx -\mathbb{E}_{\mathsf{q}(\boldsymbol{\theta})}\Big[\frac{1}{S}\sum_{s=1}^{S}\frac{1}{L^{(s)}}\ln\frac{\mathsf{P}(\boldsymbol{y}^{(s)}|\boldsymbol{x},\boldsymbol{\theta})}{\mathsf{P}(\boldsymbol{y}^{(s)}|\boldsymbol{x},\mathcal{D})}\Big], \; \boldsymbol{y}^{(s)} \sim \mathsf{P}(\boldsymbol{y}|\boldsymbol{x},\mathcal{D}) \qquad (34)$$

Similarly, we can also obtain an asymptotically exact chain-rule approximation:

$$\hat{\mathcal{M}}^{(S)}\big[\boldsymbol{y},\boldsymbol{\theta}|\boldsymbol{x},\mathcal{D}\big] = \mathbb{E}_{\mathsf{q}(\boldsymbol{\theta})}\Big[\frac{1}{L}\sum_{l=1}^{L}\mathbb{E}_{\mathsf{P}(\boldsymbol{y}_{<l}|\boldsymbol{x},\mathcal{D})}\big[\mathrm{KL}[\mathsf{P}(y_l|\boldsymbol{y}_{<l},\boldsymbol{x},\mathcal{D})\|\mathsf{P}(y_l|\boldsymbol{y}_{<l},\boldsymbol{x},\boldsymbol{\theta})]\big]\Big]$$

$$= \frac{1}{L}\sum_{l=1}^{L}\mathbb{E}_{\mathsf{P}(\boldsymbol{y}_{<l}|\boldsymbol{x},\mathcal{D})}\big[\mathcal{M}[y_l,\boldsymbol{\theta}|\boldsymbol{y}_{<l},\boldsymbol{x},\mathcal{D}]\big] \tag{35}$$

$$\approx \frac{1}{S}\sum_{s=1}^{S}\frac{1}{L^{(s)}}\sum_{l=1}^{L^{(s)}}\mathcal{M}[y_l,\boldsymbol{\theta}|\boldsymbol{y}_{<l}^{(s)},\boldsymbol{x},\mathcal{D}], \quad \forall \boldsymbol{y}_{<l}^{(s)} \subset \boldsymbol{y}^{(s)} \sim \mathsf{P}(\boldsymbol{y}|\boldsymbol{x},\mathcal{D})$$

### A.3 IMPORTANCE WEIGHTING

As discussed in section 3, beam-search decoding can be interpreted as a form of importance sampling. For the Monte-Carlo approximations for sequence-level measures of uncertainty to be used with beam search, they need to be adjusted such that uncertainty associated with each hypothesis $\boldsymbol{y}^{(b)}$ within the beam $\mathcal{B}$ is weighted in proportion to it's probability:

$$\boldsymbol{y}^{(b)} \in \mathcal{B}, \pi_b = \frac{\exp\frac{1}{T}\ln\mathsf{P}(\boldsymbol{y}^{(b)}|\boldsymbol{x},\mathcal{D})}{\sum_k^B \exp\frac{1}{T}\ln\mathsf{P}(\boldsymbol{y}^{(k)}|\boldsymbol{x},\mathcal{D})} \tag{36}$$

All chain-rule derived measures of uncertainty will be expressed as follows:

$$\hat{\mathcal{H}}_{\text{C-IW}}^{(B)}[\mathsf{P}(\boldsymbol{y}|\boldsymbol{x},\mathcal{D})] \approx \sum_{b=1}^{B}\frac{\pi_b}{L^{(b)}}\sum_{l=1}^{L^{(b)}}\mathcal{H}[\mathsf{P}(y_l|\boldsymbol{x},\boldsymbol{y}_{<l}^{(b)},\mathcal{D})] \tag{37}$$

$$\hat{\mathcal{I}}_{\text{C-IW}}^{(B)}\big[\boldsymbol{y},\boldsymbol{\theta}|\boldsymbol{x},\mathcal{D}\big] \approx \sum_{b=1}^{B}\frac{\pi_b}{L^{(b)}}\sum_{l=1}^{L^{(b)}}\mathcal{I}[y_l,\boldsymbol{\theta}|\boldsymbol{y}_{<l}^{(b)},\boldsymbol{x},\mathcal{D}] \tag{38}$$

$$\hat{\mathcal{K}}_{\text{C-IW}}^{(B)}\big[\boldsymbol{y},\boldsymbol{\theta}|\boldsymbol{x},\mathcal{D}\big] \approx \sum_{b=1}^{B}\frac{\pi_b}{L^{(b)}}\sum_{l=1}^{L^{(b)}}\mathcal{K}[y_l,\boldsymbol{\theta}|\boldsymbol{y}_{<l}^{(b)},\boldsymbol{x},\mathcal{D}] \tag{39}$$

$$\hat{\mathcal{M}}_{\text{C-IW}}^{(B)}\big[\boldsymbol{y},\boldsymbol{\theta}|\boldsymbol{x},\mathcal{D}\big] \approx \sum_{b=1}^{B}\frac{\pi_b}{L^{(b)}}\sum_{l=1}^{L^{(b)}}\mathcal{M}[y_l,\boldsymbol{\theta}|\boldsymbol{y}_{<l}^{(b)},\boldsymbol{x},\mathcal{D}] \tag{40}$$

Joint-sequence measures of uncertainty are similarly modified:

$$\hat{\mathcal{H}}_{\text{S-IW}}^{(B)}[\mathsf{P}(\boldsymbol{y}|\boldsymbol{x},\mathcal{D})] \approx -\sum_{b=1}^{B}\frac{\pi_s}{L^{(b)}}\ln\mathsf{P}(\boldsymbol{y}^{(b)}|\boldsymbol{x},\mathcal{D}) \tag{41}$$

$$\hat{\mathcal{M}}_{\text{S-IW}}^{(B)}[\boldsymbol{y},\boldsymbol{\theta}|\boldsymbol{x}] \approx \mathbb{E}_{\mathsf{q}(\boldsymbol{\theta})}\Big[\sum_{b=1}^{B}\frac{\pi_s}{L^{(b)}}\ln\frac{\mathsf{P}(\boldsymbol{y}^{(b)}|\boldsymbol{x},\mathcal{D})}{\mathsf{P}(\boldsymbol{y}^{(b)}|\boldsymbol{x},\boldsymbol{\theta})}\Big] \tag{42}$$

## B   EXPERIMENTAL CONFIGURATION

The current section of the appendix provides both a description of the datasets and details of the models and experimental setups used in this work.

### B.1   ASR MODEL CONFIGURATION

In this work ensembles of the VGG-Transformer sequence-to-sequence ASR model (Mohamed et al., 2019) were considered. An ensemble of 6 models was constructed using a different seed for both initialization and mini-batch shuffling in each model. We used ensembles of only 6 VGG-Transformer models for inference. We used the Fairseq (Ott et al., 2019) implementation and training recipe for this model with no modifications. Specifically, models were trained at a fixed learning rate for 80

Table 6: Description of ASR Datasets

| Dataset | Subset | Hours | Utterances | Words / Utterance | Domain |
|---------|--------|-------|------------|-------------------|--------|
| Librispeech | Train | 960 | 281.2K | 33.4 | |
| | Dev-Clean | 5.4 | 2703 | 17.8 | |
| | Dev-Other | 5.3 | 2864 | 18.9 | |
| | Test-Clean | 5.4 | 2620 | 20.1 | Story Books |
| | Test-Other | 5.1 | 2939 | 17.8 | |
| AMI | Eval | - | 12643 | 7.1 | Meetings |
| Common-Voice FR | Test | - | 14760 | 9.5 | General |

epochs, where an epoch is a full pass through the entire training set. Checkpoints over the last 30 epochs were averaged together, which proved to be crucial to ensuring good performance. Training took 8 days using 8 V100 GPUs. Models were trained on the full 960 hours of the LibriSpeech dataset (Panayotov et al., 2015) in exactly the same configuration as described in (Mohamed et al., 2019). LibriSpeech is a dataset with ~1000 hours of read books encoded in 16-bit, 16kHz FLAC format. The reference transcriptions were tokenized using a vocabulary of 5000 tokens, as per the standard recipe in Fairseq for the VGG-transformer (Ott et al., 2019; Mohamed et al., 2019). For OOD detection we considered the evaluation subset of the AMI dataset (Kraaij et al., 2005), which is a dataset of meeting transcriptions, as well as the Russian and French datasets of the Common Voice Project (Ardila et al., 2019), which consist of people reading diverse text from the internet. AMI is encoded in 16-bit, 16Khz WAV format. Common Voice data was stored as 24kHz 32-bit MP3 files which were converted into 16-bit 16kHz WAV format via the SOX tool. WER was evaluated using the NIST SCLITE scoring tool.

## B.2 NMT MODEL CONFIGURATION

Table 7: Description of NMT Datasets

| Dataset | Subset | LNG | Sentences | Words / Sent. | Domain |
|---------|--------|-----|-----------|---------------|--------|
| WMT'14 EN-FR | Train | En | 40.8M | 29.2 | |
| | | Fr | | 33.5 | Policy, News, Web |
| WMT'17 EN-DE | Train | En | 4.5M | 26.2 | |
| | | De | | 24.8 | Policy, News, Web |
| Newstest14 | - | En | 3003 | 27.0 | |
| | - | Fr | | 32.1 | News |
| | - | De | | 28.2 | |
| Khresmoi-Summary | Dev+Test | En | 1500 | 19.0 | |
| | | Fr | | 21.8 | Medical |
| | | De | | 17.9 | |

This work considered ensembles of Transformer-Big (Vaswani et al., 2017) neural machine translation (NMT) models. An ensemble 10 models was constructed using a different seed for both initialization and mini-batch shuffling in each model. NMT models were trained on the WMT'14 English-French and WMT'17 English-German datasets. All models were trained using the standard Fairseq (Ott et al., 2019) implementation and recipe, which is consistent with the baseline setup in described in (Ott et al., 2018b). The data was tokenized using a BPE vocabulary of 40,000 tokens as per the standard recipe (Sennrich et al., 2015). For each dataset and translation direction an ensemble of 10 models was trained using different random seeds. All 10 models were used during inference. Models trained on WMT'17 English-German were trained for 193000 steps of gradient descent, which corresponds to roughly 49 epochs, while WMT'14 English-French models were trained for 800000 steps of gradient descent, which corresponds to roughly 19 epochs. Models were checkpoint-averaged across the last 10 epochs. All models were trained using mixed-precision training. Models were evaluated on newstest14, which was treated as in-domain data. OOD data was constructed by considering

BPE-token permuted and language-flipped versions of the newstest14 dataset. Furthermore, the *khresmoi-summary* medical dataset as well the reference transcriptions of the LibriSpeech test-clean and test-other datasets were also used as OOD evaluation datasets. All additional datasets used consistent tokenization using the 40K BPE vocabulary.

## C  PREDICTIVE PERFORMANCE ABLATION STUDIES

The current section provides additional results assessing the predictive-performance and negative log-likelihood of ensembles of autoregressive NMT and ASR models. Additionally, we include an ablation study of how the number of models in an ensemble affects the performance in terms of BLEU and NLL. Tables 8 and 9 include expanded set of results. Crucially the results show that for all languages, tasks and datasets a *product-of-expectations* yields superior performance (with one exception) in beam-search decoding and and a consistently lower NLL on reference transcriptions and translations.

Table 8: Predictive performance in terms of BLEU, %WER and NLL on newstest14 and LibriSpeech.

| Model | NWT'14 BLEU | | MED BLEU | | NWT'14 NLL | | MED NLL | |
|---|---|---|---|---|---|---|---|---|
| | EN-DE | EN-FR | EN-DE | EN-FR | EN-DE | EN-FR | EN-DE | EN-FR |
| Single | 28.8 $_{\pm 0.2}$ | 45.4 $_{\pm 0.3}$ | 29.9 $_{\pm 0.5}$ | 51.3 $_{\pm 0.5}$ | 1.46 $_{\pm 0.02}$ | 1.10 $_{\pm 0.01}$ | 1.29 $_{\pm 0.01}$ | 0.82 $_{\pm 0.01}$ |
| ENS-PrEx | **30.1** | **46.5** | **32.1** | 52.7 | **1.33** | **1.04** | **1.16** | **0.77** |
| ENS-ExPr | 29.9 | 46.3 | 31.4 | **52.8** | 1.36 | 1.05 | 1.19 | 0.78 |

Table 9: Predictive performance in terms of BLEU, %WER and NLL on newstest14 and LibriSpeech.

| Model | ASR % WER | | | ASR NLL | | |
|---|---|---|---|---|---|---|
| | LTC | LTO | AMI | LTC | LTO | AMI |
| Single | 5.6 $_{\pm 0.2}$ | 14.7 $_{\pm 0.5}$ | 78.7 $_{\pm 13.4}$ | 0.34 $_{\pm 0.00}$ | 0.86 $_{\pm 0.02}$ | 5.78 $_{\pm 0.32}$ |
| ENS-PrEx | **4.2** | **11.3** | **50.4** | **0.20** | **0.48** | **4.05** |
| ENS-ExPr | 4.5 | 12.6 | 53.4 | 0.23 | 0.58 | 4.62 |

Finally, we present an ablation study, which shows how the predictive performance varies with the number of models in an ensemble. The ablation shows several trends. Firstly, both BLEU/WER and NLL begin to shown diminishing returns for using more models. This suggests that using 4-6 NMT models and 2-3 ASR models will allow most of the gains to be derived at half the cost of a full 10 or 6-model ensemble. Secondly, it shows that the advantage of a product-of-expectations combination is remains consistent with the number of models. This shows that regardless of the number of models available, it is always better to combine as a product-of-expectations.

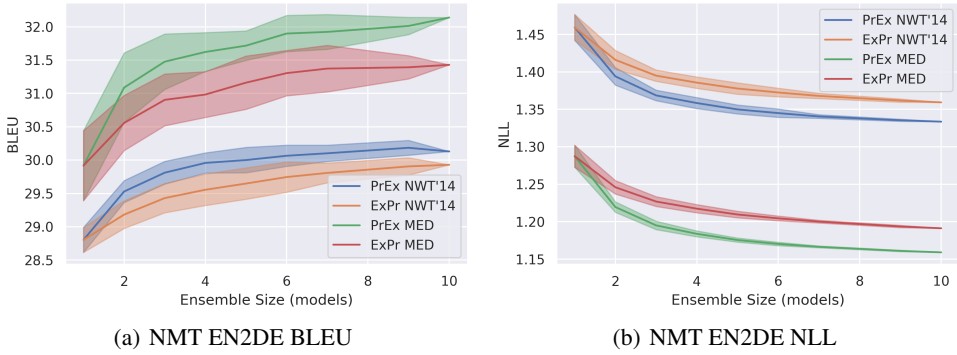

(a) NMT EN2DE BLEU  (b) NMT EN2DE NLL

Figure 2: BLEU and NLL ablation study. Shading indicates $\pm 2\sigma$

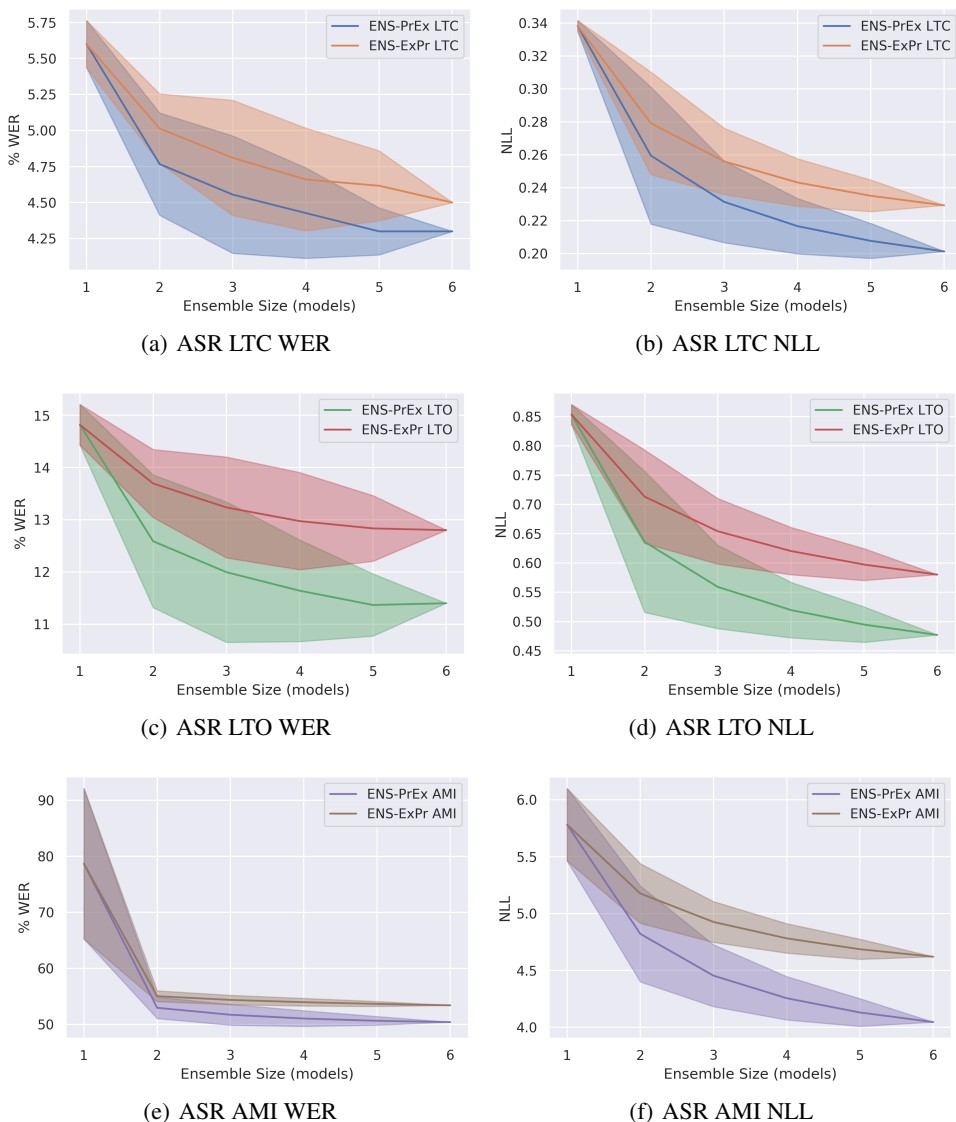

Figure 3: WER and NLL ablation study. Shading indicates $\pm 2\sigma$

## D    SEQUENCE-LEVEL ERROR DETECTION

The current appendix provides a description of the Prediction Rejection Ratio metric, the rejection curves which correspond to results in section 4, and histograms of sentence-WER and sentence-BLEU which provide insights into the behaviour of the corresponding rejection curves.

### D.1    PREDICTION REJECTION RATIO

Here we describe the *Prediction Rejection Ratio* metric, proposed in (Malinin, 2019; Malinin et al., 2017), which in this work is used to assess how well measures of sequence-level uncertainty are able to identify sentences which are hard to translate/transcribe. Consider the task of identifying misclassifications - ideally we would like to detect all of the inputs which the model has misclassified based on a measure of uncertainty. Then, the model can either choose to not provide any prediction for these inputs, or they can be passed over or 'rejected' to an oracle (ie: human) to obtain the correct prediction (or translation/transcription). The latter process can be visualized using a *rejection curve* depicted in figure 4, where the predictions of the model are replaced with predictions provided by an

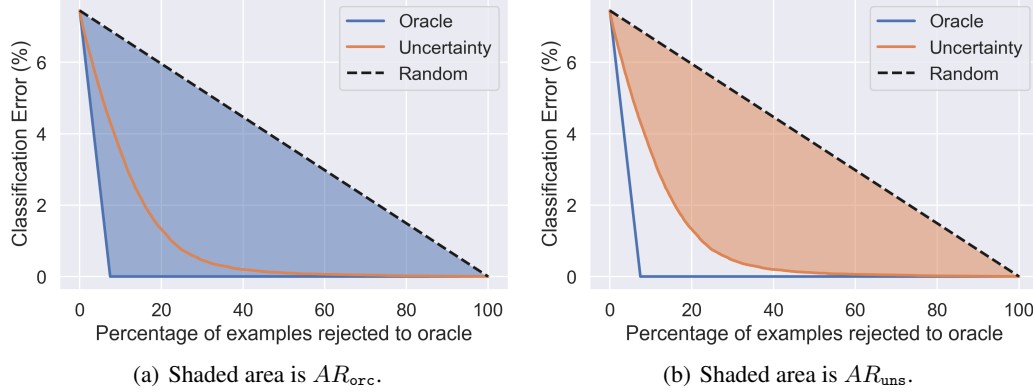

(a) Shaded area is $AR_{\mathrm{orc}}$.

(b) Shaded area is $AR_{\mathrm{uns}}$.

Figure 4: Example Prediction Rejection Curves (Malinin, 2019)

oracle in some particular order based on estimates of uncertainty. If the estimates of uncertainty are uninformative, then, in expectation, the rejection curve would be a straight line from base error rate to the lower right corner, given the error metric is a linear function of individual errors. However, if the estimates of uncertainty are 'perfect' and always bigger for a misclassification than for a correct classification, then they would produce the 'oracle' rejection curve. The 'oracle' curve will go down linearly to $0\%$ classification error at the percentage of rejected examples equal to the number of misclassifications. A rejection curve produced by estimates of uncertainty which are not perfect, but still informative, will sit between the 'random' and 'oracle' curves. The quality of the rejection curve can be assessed by considering the *ratio* of the area between the 'uncertainty' and 'random' curves $AR_{\mathrm{uns}}$ (orange in figure 4) and the area between the 'oracle' and 'random' curves $AR_{\mathrm{orc}}$ (blue in figure 4). This yields the *prediction rejection area ratio PRR*:

$$PRR = \frac{AR_{\mathrm{uns}}}{AR_{\mathrm{orc}}} \qquad (43)$$

A rejection area ratio of 1.0 indicates optimal rejection, a ratio of 0.0 indicates 'random' rejection. A negative rejection ratio indicates that the estimates of uncertainty are 'perverse' - they are higher for accurate predictions than for misclassifications. An important property of this performance metric is that it is independent of classification performance, unlike AUPR, and thus it is possible to compare models with different base error rates. Note, that similar approaches to assessing misclassification detection were considered in (Lakshminarayanan et al., 2017; Malinin et al., 2017; Malinin, 2019). In this work instead of considered misclassifications we assess whether measures of uncertainty correlate well with sentence-level BLEU or WER. The overall 'error' is then the average of sentence-level BLEU/WER over the test-set.

## D.2 Rejection Curves

The rejection curves for all NMT models on newstest14 and the ASR model on LibriSpeech test-clean and test-other are presented in figure 5. The main difference between the NMT and ASR curves is that the 'oracle' rejection curve for the former is not much better than random, while the rejection curve for the latter is far better than random. This can be explained by considering the histograms of sentence-level BLEU and sentence-level WER presented in figure 6. Notice, that the sentence-level BLEUs are varied across the spectrum, and very few sentences reach a BLEU of 100. In contrast, 55-75% of all utterances transcribed by the ASR models have a sentence-WER of 0-10%, and then there are a few utterances with a much larger WER. Thus, if the measures of uncertainty can identify the largest errors, which contribute most to the mean WER over the dataset, then a large decrease can be achieved. Hence the shape of the 'oracle' WER-rejection curve. In contrast, the contributions from each sentence to mean sentence-BLEU are more evenly spread. Thus, it is difficult to significantly raise the mean-sentence BLEU by rejecting just a few sentences. Hence the shape of the 'oracle' BLEU rejection curve for NMT.

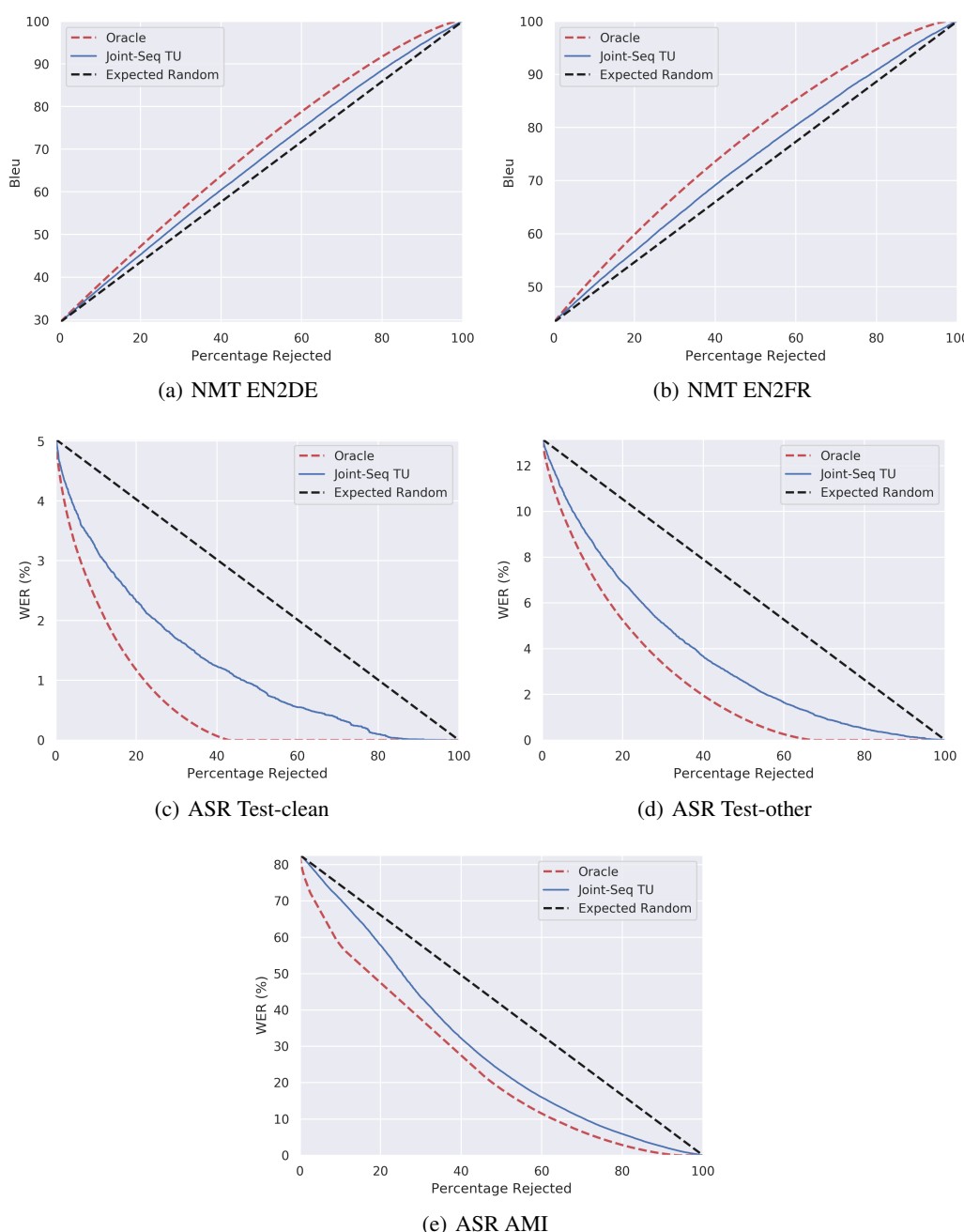

Figure 5: Sequence-level rejection curves for NMT and ASR.

Figure 6e shows that the sentence-WER on AMI eval is distributed more like the sentence-BLEU is for NMT tasks - few correct sentence and a much more uniform distribution of error. Thus, the corresponding 'oracle' rejection curve's shape is more similar to the NMT 'oracle' rejection curves. This clearly shows that the shape of the oracle curve is not determined by the task (ASR/NMT), but the error (BLEU/WER) distribution across a dataset.

The second trend in the results provided in section 4 is that score-based measures of uncertainty work better than entropy-based measures on NMT tasks, while on ASR they perform comparably. The justification provided states that NMT models yield far less confident predictions, and therefore entropy-based measures suffer due to probability mass assigned to other tokens. In contrast, ASR

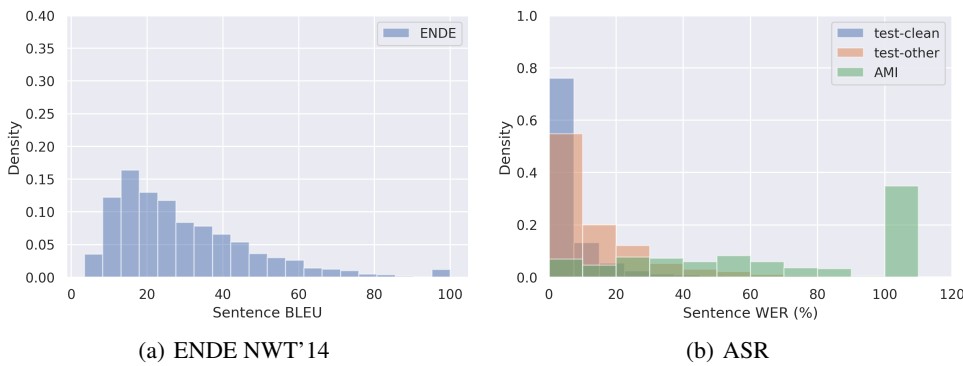

(a) ENDE NWT'14  (b) ASR

Figure 6: Sentence BLEU and WER Histograms.

models yield more confident predictions, as shown in figure 7. Notably, on AMI and Common Voice datasets the ASR model also yields less confident predictions, and thus the score-based measures of uncertainty do better than entropy-based ones in the AMI rejection curve in figure 5e. These results show that on tasks where it is important to determine which particular translation/transcription hypotheses are worse, score-based measures of uncertainty do as well as or better than entropy-based measures. This result is consistent with confidences being a better measure of uncertainty for misclassification detection in unstructured prediction tasks (Malinin, 2019).

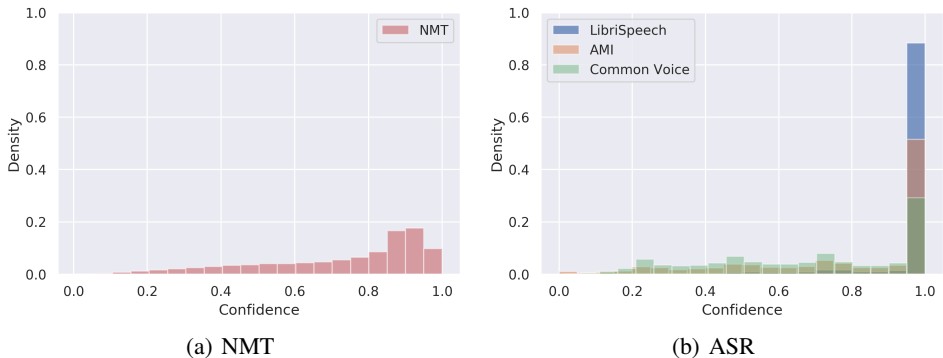

(a) NMT  (b) ASR

Figure 7: Histograms of predicted-token confidence for ASR and NMT.

### D.3 Additional Results

The current section provides additional sequence-level error detection results. We examine sequence-level error detection of hypotheses produced by an ensemble combined as an *expectation-of-products*. Results presented in table 10 serve to confirm the previously observed trends and illustrate that the superior performance of measures of uncertainty derived from an *product-of-expectations* posterior does not depend on the nature of the hypotheses.

## E   Token-level Error Detection

Current appendix provides additional results for token-level error detection. Notably, we present results using a score-based measures of *knowledge uncertainty* $\mathcal{M}_{\omega_l}$, as well as results on hypotheses derived from an expectation-of-products ensemble combination. Results in table 11 that the new measures of uncertainty consistently outperforms token-level mutual-information and RMI.

Results in table 12 show that the observed trends do not depend from which ensemble-combination the hypotheses were obtained.

Table 10: Sequence-level Error Detection % PRR in Beam-Search decoding using $P_{EP}(\boldsymbol{y}|\boldsymbol{x}, \mathcal{D})$.

| Task | Test set | ENS-PrEx TU | | ENS-ExPr TU | | | | ENS-PrEx KU | | | | | ENS-ExPr KU | | |
| | | $\hat{\mathcal{H}}_{\text{C-IW}}^{(1)}$ | $\hat{\mathcal{H}}_{\text{S-IW}}^{(1)}$ | $\hat{\mathcal{H}}_{\text{C-IW}}^{(1)}$ | $\hat{\mathcal{H}}_{\text{S-IW}}^{(B)}$ | $\hat{\mathcal{I}}_{\text{C-IW}}^{(1)}$ | $\hat{\mathcal{K}}_{\text{C-IW}}^{(1)}$ | $\hat{\mathcal{M}}_{\text{C-IW}}^{(1)}$ | $\hat{\mathcal{M}}_{\text{S-IW}}^{(1)}$ | $\hat{\mathcal{I}}_{\text{C-IW}}^{(1)}$ | $\hat{\mathcal{K}}_{\text{C-IW}}^{(1)}$ | $\hat{\mathcal{M}}_{\text{C-IW}}^{(B)}$ | $\hat{\mathcal{M}}_{\text{S-IW}}^{(1)}$ |
|---|---|---|---|---|---|---|---|---|---|---|---|---|---|
| LSP | LTC | 64.7 | **68.5** | 63.5 | 67.3 | 62.8 | 61.1 | 60.5 | 64.3 | 52.8 | 60.6 | 60.3 | 64.2 |
| | LTO | 73.5 | **76.0** | 68.8 | 72.0 | 72.3 | 69.5 | 68.6 | 70.5 | 36.0 | 68.3 | 68.3 | 70.9 |
| | AMI | 57.9 | **66.8** | 54.7 | 61.0 | 53.5 | 51.2 | 50.1 | 63.1 | 16.7 | 55.2 | 54.0 | 62.8 |
| ENDE | nwt14 | 29.9 | **46.6** | 28.6 | 45.6 | 29.1 | 28.1 | 27.4 | 30.4 | 5.6 | 22.4 | 28.1 | 30.6 |
| | MED | 31.1 | **45.5** | 28.9 | 41.7 | 37.1 | 36.4 | 35.8 | 38.9 | -2.1 | 26.8 | 36.7 | 38.6 |
| ENFR | nwt14 | 26.3 | **38.7** | 25.9 | 38.4 | 31.3 | 30.8 | 30.3 | 32.3 | 16.9 | 27.6 | 30.1 | 30.2 |
| | MED | 17.3 | **40.1** | 16.4 | 39.2 | 35.3 | 35.1 | 34.9 | 39.9 | 14.1 | 29.2 | 35.4 | 37.1 |

Table 11: %AUPR for LibriSpeech in Beam-Search Decoding regime using $P_{PE}(\boldsymbol{y}|\boldsymbol{x}, \mathcal{D})$.

| Test Data | ENSM-PrEx TU | | ENSM-ExPr TU | | ENS-PrEx KU | | | | ENS-ExPr KU | | | | % TER |
| | $\mathcal{H}$ | $\mathcal{P}$ | $\mathcal{H}$ | $\mathcal{P}$ | $\mathcal{I}$ | $\mathcal{K}$ | $\mathcal{M}$ | $\mathcal{M}_{\omega_l}$ | $\mathcal{I}$ | $\mathcal{K}$ | $\mathcal{M}$ | $\mathcal{M}_{\omega_l}$ | |
|---|---|---|---|---|---|---|---|---|---|---|---|---|---|
| LTC | 34.7 | **36.3** | 32.4 | 33.4 | 32.7 | 28.0 | 27.6 | 33.4 | 26.5 | 25.9 | 27.4 | 30.8 | 3.8 |
| LTO | 42.4 | **43.3** | 39.0 | 39.1 | 40.9 | 37.1 | 36.1 | 41.5 | 30.8 | 33.6 | 35.3 | 37.4 | 10.2 |
| AMI | 71.7 | **74.6** | 68.3 | 70.4 | 71.8 | 68.7 | 67.9 | 72.3 | 59.2 | 64.4 | 66.6 | 67.3 | 41.2 |

# F    OUT-OF-DISTRIBUTION INPUT DETECTION

In the current section additional OOD input detection results are provided for En-De and En-Fr NMT models and the ASR model in a Beam-Search decoding regime. Additionally, we provide results for En-De and En-Fr models on reference hypotheses in a teacher-forcing regime.

## F.1    ADDITIONAL RESULTS

Table 13 provides a full of OOD detection results using an ensemble of EN-FR translation models. All hypotheses are derived from a product-of-expectations ensemble. These results tell essentially the same story as OOD detection on En-De models. However, it seem that because WMT'14 En-Fr is roughly ten times larger than WMT'17 En-Fr, OOD detection in some cases, notably LTC, PRM and L-FR is easier. However, L-DE are significantly worse. Note that for En-FR, L-DE is the heldout language, while L-FR is more familiar. One explanation is that the copy-through effect is so strong on an unfamiliar language that even measures of *knowledge uncertainty* are drastically affected. This suggests that it is necessary to eliminate this regime, as it strongly compromises the measures of uncertainty.

Table 14 provides a full set of OOD detection results on hypotheses generated from an ensemble combined as an *expectation-of-products*. The results essentially tell the same story those obtained on hypotheses generated from an ensemble combined as a *product-of-expectations*. This confirms the trend that measures of uncertainty derived by expressing the predictive posterior as a *product-of-expectations* typically yield marginally better performance, with expceptions where they yield significantly better performance, regardless of the nature of the hypotheses.

## F.2    TEACHER-FORCING

Table 15 provides a set of OOD detection results for EN-DE/EN-FR translation models evaluated in a teacher-forcing regime, where 'references' are fed into the decoder. The aim is to further explore

Table 12: %AUPR for LibriSpeech in Beam-Search Decoding regime using $P_{EP}(\boldsymbol{y}|\boldsymbol{x}, \mathcal{D})$.

| Test Data | ENSM-PrEx TU | | ENSM-ExPr TU | | ENS-PrEx KU | | | | ENS-ExPr KU | | | | % TER |
| | $\mathcal{H}$ | $\mathcal{P}$ | $\mathcal{H}$ | $\mathcal{P}$ | $\mathcal{I}$ | $\mathcal{K}$ | $\mathcal{M}$ | $\mathcal{M}_{\omega_l}$ | $\mathcal{I}$ | $\mathcal{K}$ | $\mathcal{M}$ | $\mathcal{M}_{\omega_l}$ | |
|---|---|---|---|---|---|---|---|---|---|---|---|---|---|
| LTC | 38.0 | **43.2** | 34.1 | 37.2 | 37.0 | 32.9 | 32.0 | 40.5 | 27.7 | 32.1 | 33.9 | 39.6 | 4.1 |
| LTO | 50.8 | **56.4** | 44.5 | 46.5 | 50.3 | 46.2 | 45.2 | 54.1 | 32.9 | 45.1 | 48.1 | 53.5 | 12.0 |
| AMI | 76.5 | **80.1** | 72.1 | 73.9 | 76.9 | 74.0 | 73.2 | 77.9 | 59.5 | 73.4 | 75.5 | 77.6 | 44.1 |

Table 13: OOD Detection % ROC-AUC in Beam-Search decoding regime for ASR and NMT. $T = 10$

| Task | OOD Data | B | ENS-PrEx TU $\hat{\mathcal{H}}_{C\text{-}IW}^{(B)}$ | $\hat{\mathcal{H}}_{S\text{-}IW}^{(B)}$ | ENS-ExPr TU $\hat{\mathcal{H}}_{C\text{-}IW}^{(B)}$ | $\hat{\mathcal{H}}_{S\text{-}IW}^{(B)}$ | ENS-PrEx KU $\hat{\mathcal{T}}_{C\text{-}IW}^{(B)}$ | $\hat{\mathcal{K}}_{C\text{-}IW}^{(B)}$ | $\hat{\mathcal{M}}_{C\text{-}IW}^{(B)}$ | $\hat{\mathcal{M}}_{S\text{-}IW}^{(B)}$ | ENS-ExPr KU $\hat{\mathcal{T}}_{C\text{-}IW}^{(B)}$ | $\hat{\mathcal{K}}_{C\text{-}IW}^{(B)}$ | $\hat{\mathcal{M}}_{C\text{-}IW}^{(B)}$ | $\hat{\mathcal{M}}_{S\text{-}IW}^{(B)}$ |
|---|---|---|---|---|---|---|---|---|---|---|---|---|---|---|
| ENFR | LTC | 1 | 64.1 | 77.3 | 63.3 | 77.2 | 78.5 | 78.4 | 78.3 | 81.7 | 65.2 | 75.3 | 78.0 | 78.9 |
|  |  | 5 | 65.4 | 79.2 | 64.7 | 79.2 | 79.1 | 78.9 | 78.8 | 83.6 | 66.7 | 76.8 | 78.6 | 82.0 |
|  | PRM | 1 | 92.7 | 91.6 | 90.9 | 91.6 | **98.7** | 98.6 | 98.6 | 98.5 | 61.3 | 94.0 | **98.6** | 97.7 |
|  |  | 5 | 93.3 | 92.1 | 91.7 | 92.7 | **98.7** | **98.7** | 98.6 | **98.7** | 62.6 | 95.1 | **98.7** | 98.2 |
|  | L-FR | 1 | 55.2 | 33.4 | 51.5 | 35.7 | 86.6 | 88.0 | 88.9 | 84.8 | 58.2 | 82.6 | 88.3 | 85.6 |
|  |  | 5 | 57.2 | 42.1 | 53.4 | 46.0 | 88.1 | 89.6 | 90.4 | **94.8** | 60.1 | 85.9 | 90.3 | 94.4 |
|  | L-DE | 1 | 12.4 | 6.9 | 11.2 | 7.6 | 35.1 | 38.2 | 40.4 | 39.2 | 19.7 | 27.6 | 40.1 | 44.4 |
|  |  | 5 | 13.1 | 14.5 | 11.8 | 15.6 | 38.9 | 42.7 | 45.4 | **67.8** | 19.4 | 32.0 | 46.6 | 67.6 |

Table 14: OOD Detection % ROC-AUC in Beam-Search decoding regime for ASR and NMT.

| Task | OOD Data | T | B | ENS-PrEx TU $\hat{\mathcal{H}}_{C\text{-}IW}^{(B)}$ | $\hat{\mathcal{H}}_{S\text{-}IW}^{(B)}$ | ENS-ExPr TU $\hat{\mathcal{H}}_{C\text{-}IW}^{(B)}$ | $\hat{\mathcal{H}}_{S\text{-}IW}^{(B)}$ | ENS-PrEx KU $\hat{\mathcal{T}}_{C\text{-}IW}^{(B)}$ | $\hat{\mathcal{K}}_{C\text{-}IW}^{(B)}$ | $\hat{\mathcal{M}}_{C\text{-}IW}^{(B)}$ | $\hat{\mathcal{M}}_{S\text{-}IW}^{(B)}$ | ENS-ExPr KU $\hat{\mathcal{T}}_{C\text{-}IW}^{(B)}$ | $\hat{\mathcal{K}}_{C\text{-}IW}^{(B)}$ | $\hat{\mathcal{M}}_{C\text{-}IW}^{(B)}$ | $\hat{\mathcal{M}}_{S\text{-}IW}^{(B)}$ |
|---|---|---|---|---|---|---|---|---|---|---|---|---|---|---|---|
| ASR | LTO | 1 | 1 | 76.5 | 75.4 | 75.6 | 74.6 | 76.2 | 76.5 | 76.4 | 73.9 | 68.5 | 76.0 | 76.1 | 74.1 |
|  |  |  | 20 | 77.0 | 77.0 | 76.3 | 75.5 | 76.8 | 77.1 | 77.1 | 77.0 | 71.8 | 76.9 | 77.0 | 77.2 |
|  | AMI | 1 | 1 | 97.5 | 97.4 | 97.3 | 97.1 | 96.3 | 96.2 | 96.1 | 96.2 | 79.7 | 96.1 | 96.0 | 96.2 |
|  |  |  | 20 | 96.6 | 97.9 | 96.7 | 97.3 | 95.1 | 95.1 | 95.0 | 97.7 | 86.7 | 95.2 | 95.2 | 97.7 |
|  | C-FR | 1 | 1 | 99.9 | 99.7 | 99.7 | 99.1 | 99.9 | 99.9 | 99.9 | 99.8 | 33.4 | 99.9 | 99.9 | 99.9 |
|  |  |  | 20 | 100.0 | 99.7 | 99.9 | 98.9 | 99.9 | 99.9 | 99.9 | 99.9 | 38.9 | 99.9 | 99.9 | 99.9 |
| ENDE | LTC | 10 | 1 | 65.3 | 72.0 | 63.7 | 70.1 | 72.3 | 72.0 | 71.7 | 72.8 | 49.5 | 66.4 | 71.0 | 71.2 |
|  |  |  | 5 | 65.4 | 74.8 | 63.8 | 72.3 | 72.7 | 72.4 | 72.1 | 75.0 | 50.6 | 67.4 | 71.4 | 73.9 |
|  | PRM | 10 | 1 | 83.0 | 85.2 | 78.6 | 79.4 | 96.4 | 96.7 | 96.7 | 95.9 | 45.3 | 92.3 | 96.3 | 94.3 |
|  |  |  | 5 | 83.6 | 86.3 | 79.3 | 79.5 | 96.7 | 96.9 | 97.0 | 96.5 | 47.6 | 93.5 | 96.6 | 95.6 |
|  | L-FR | 10 | 1 | 27.1 | 20.4 | 21.8 | 16.5 | 63.3 | 68.9 | 72.2 | 69.9 | 19.8 | 43.7 | 69.4 | 69.4 |
|  |  |  | 5 | 28.0 | 23.9 | 22.7 | 20.2 | 65.9 | 71.7 | 75.2 | 78.6 | 18.9 | 47.5 | 73.4 | 76.8 |
|  | L-DE | 10 | 1 | 41.7 | 33.0 | 34.6 | 24.4 | 74.9 | 78.5 | 80.5 | 75.2 | 34.1 | 67.8 | 78.3 | 73.2 |
|  |  |  | 5 | 42.6 | 39.8 | 35.7 | 31.2 | 76.9 | 80.6 | 82.6 | 88.6 | 33.3 | 72.2 | 81.2 | 85.5 |
| ENFR | LTC | 10 | 1 | 63.1 | 78.0 | 61.6 | 76.5 | 78.1 | 78.0 | 77.9 | 81.6 | 56.2 | 73.0 | 77.5 | 79.2 |
|  |  |  | 5 | 64.4 | 80.1 | 63.1 | 78.3 | 78.7 | 78.6 | 78.5 | 83.8 | 58.7 | 75.0 | 78.2 | 82.2 |
|  | PRM | 10 | 1 | 93.2 | 92.7 | 90.7 | 90.6 | 98.7 | 98.7 | 98.6 | 98.5 | 38.2 | 88.8 | 98.6 | 98.1 |
|  |  |  | 5 | 93.7 | 93.7 | 91.4 | 91.2 | 98.8 | 98.7 | 98.7 | 98.8 | 38.6 | 90.6 | 98.8 | 98.7 |
|  | L-FR | 10 | 1 | 55.9 | 38.3 | 49.9 | 29.5 | 86.6 | 88.0 | 88.9 | 84.6 | 45.2 | 81.2 | 88.1 | 83.9 |
|  |  |  | 5 | 58.1 | 48.0 | 52.2 | 38.8 | 88.3 | 89.7 | 90.5 | 94.8 | 46.7 | 84.8 | 90.2 | 93.4 |
|  | L-DE | 10 | 1 | 12.5 | 7.6 | 11.0 | 6.0 | 35.0 | 38.1 | 40.3 | 38.9 | 20.6 | 27.6 | 39.8 | 41.3 |
|  |  |  | 5 | 13.2 | 15.4 | 11.6 | 13.6 | 39.2 | 43.1 | 45.8 | 67.6 | 18.6 | 30.8 | 46.8 | 66.6 |

Table 15: OOD Detection % ROC-AUC in Teacher-Forcing regime

| Task | OOD Data | ENS-PrEx TU $\hat{\mathcal{H}}_{C\text{-}IW}^{(1)}$ | $\hat{\mathcal{H}}_{S\text{-}IW}^{(1)}$ | ENS-ExPr TU $\hat{\mathcal{H}}_{C\text{-}IW}^{(1)}$ | $\hat{\mathcal{H}}_{S\text{-}IW}^{(1)}$ | ENS-PrEx KU $\hat{\mathcal{T}}_{C\text{-}IW}^{(1)}$ | $\hat{\mathcal{K}}_{C\text{-}IW}^{(1)}$ | $\hat{\mathcal{M}}_{C\text{-}IW}^{(1)}$ | $\hat{\mathcal{M}}_{S\text{-}IW}^{(1)}$ | ENS-ExPr KU $\hat{\mathcal{T}}_{C\text{-}IW}^{(1)}$ | $\hat{\mathcal{K}}_{C\text{-}IW}^{(1)}$ | $\hat{\mathcal{M}}_{C\text{-}IW}^{(1)}$ | $\hat{\mathcal{M}}_{S\text{-}IW}^{(1)}$ |
|---|---|---|---|---|---|---|---|---|---|---|---|---|---|
| ENDE | L-DEEN | 98.3 | 98.9 | 98.0 | 99.0 | 99.4 | **99.5** | **99.5** | 99.4 | 55.2 | 97.6 | **99.5** | 96.7 |
|  | L-DEDE | 22.4 | 11.3 | 18.6 | 11.2 | 58.7 | 64.0 | **67.1** | 41.0 | 41.3 | 54.7 | 61.4 | 48.7 |
|  | L-ENEN | 62.2 | 58.3 | 56.0 | 56.7 | 79.6 | 81.6 | 82.7 | 76.6 | 31.2 | 59.6 | 79.9 | **82.9** |
|  | L-FREN | **100.0** | 99.8 | **100.0** | 99.8 | 98.1 | 97.2 | 96.2 | 94.1 | 71.6 | 94.6 | 96.1 | 85.6 |
| ENFR | L-FREN | 99.6 | 98.4 | 99.5 | 98.6 | **99.9** | **99.9** | 99.8 | 99.6 | 67.6 | 98.7 | **99.9** | 97.2 |
|  | L-FRFR | 28.8 | 12.3 | 25.7 | 12.7 | 75.1 | 78.2 | **80.0** | 58.9 | 55.5 | 72.2 | 76.9 | 61.5 |
|  | L-ENEN | 39.4 | 40.5 | 37.7 | 40.7 | 84.3 | 86.2 | **87.4** | 54.2 | 68.1 | 82.2 | 84.8 | 56.2 |
|  | L-DEEN | **100.0** | 99.9 | **100.0** | 99.9 | 97.6 | 97.0 | 96.4 | 95.3 | 76.2 | 95.2 | 96.7 | 88.0 |

OOD detection of foreign languages and the copy-through effect. Here, in-domain data is En-De and En-Fr newstest14 for En-De and En-Fr models, respectively. As OOD data we also consider newstest14 data, but where the either source, target or both languages are changed. The results show that when the source and target languages are *both* changed (L-DEEN, L-FREN), then this is an easy to detect scenario, as copy-though is forcibly avoided. We also consider situations where we forcibly initiate copy-through. Here, we have matched pairs of source-source or target-target language. Measures of *total uncertainty* fail, while measures of *knowledge uncertainty* do not. However, the effect is more severe when source sentences are copied through, rather than target sentences. We speculate that this is an affect of the decoder being familiar with target sentences and still trying to do something sensible. These results clearly show that if the copy-through effect is somehow eliminated, then detection of OOD sentences by NMT models because as easy as for ASR models, where copy-though cannot occur by construction. Finally, we note that again, deriving uncertainties from a *product-of-expectations* predictive posterior yields marginally better results. Additionally, chain-rule RMI seems to be the best measures of *knowledge uncertainty* overall. Note that in teacher-forcing, we have access to only a *single* hypothesis per input, which is a regime where joint-sequence estimates of knowledge uncertainty, specifically RMI, tend to perform worse than chain-rule derived estimates.

## G  SENSITIVITY OF MC ESTIMATORS TO NUMBER OF SAMPLES AND CALIBRATION

In this section we explore in detail the effect of using more than the 1-best hypothesis in the Monte-Carlo estimation of entropy and reverse mutual information (RMI). We consider both chain-rule and joint-sequence estimators. Performance is evaluated on the tasks of sequence error detection and OOD detection. Figure 8 shows that for sequence error detection using more hypotheses within the beam either has little effect, or is detrimental. This is expected, as this makes the estimate less sensitive to the particular hypothesis which is being assessed. Notable, joint-sequence estimates are affected the most, while chain-rule estimators demonstrate more stable behaviour.

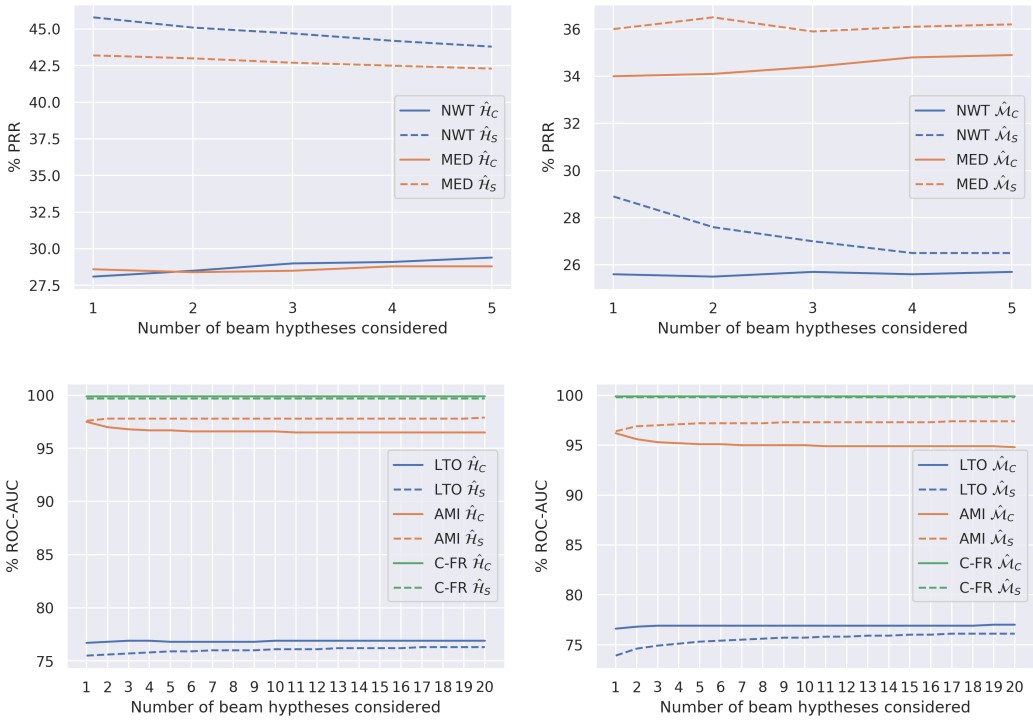

Figure 8: Sensitivity of uncertainty measures to number samples on NMT and ASR sequence error detection.

Figure 8 shows that considering more hypotheses is generally beneficial for OOD detection, especially for joint-sequence estimates of RMI. This is also expected, as for OOD detection it is useful to have information about the effect of the input on more than just the 1-best hypothesis. The performance gain is, ultimately, unsatisfying.

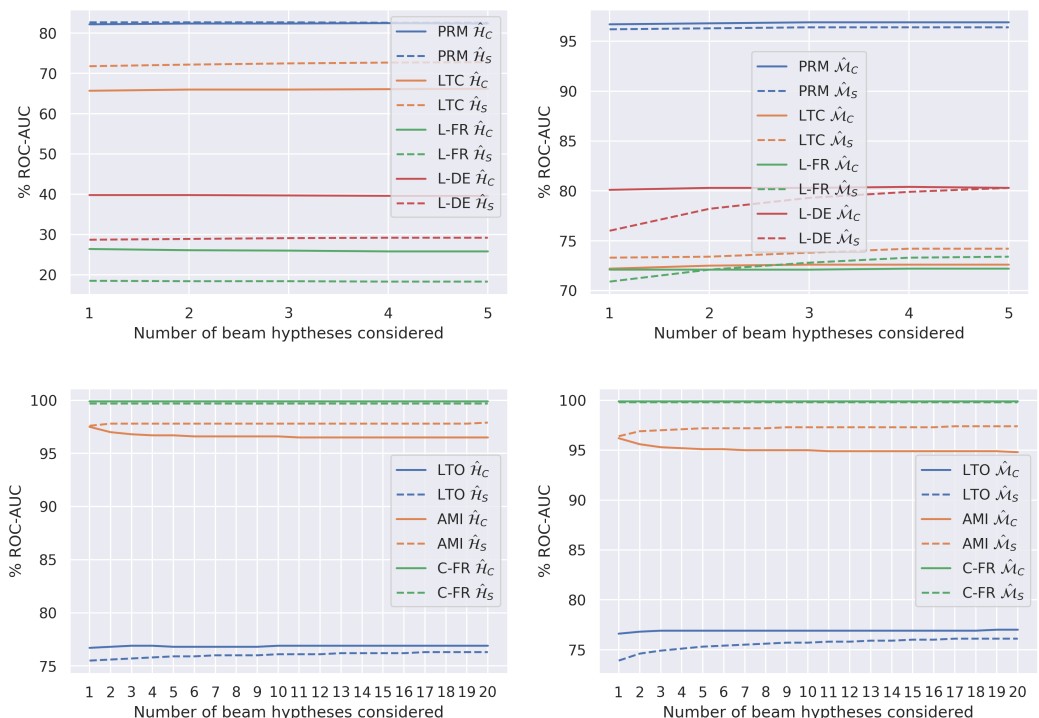

Figure 9: Sensitivity of uncertainty measures to number samples on NMT and ASR OOD detection.

We consider what happens if we increase the importance weighting temperature T, which increases the contribution for the remaining hypotheses to the uncertainty estimate. Figure 10 shows, unsurprisingly that this is extremely detrimental to sequence error detection, as it introduce even more irrelevant information. However, figure 11 shows that for OOD detection, the is unexpectedly interesting behaviour. Using higher temperature for NMT always leads to significantly performance, especially for joint-sequence estimates of RMI. However, what is especially surprising, is that for ASR the OOD detection performance degrades. It is not entirely clear why up-weighting information from competing hypotheses is detrimental to performance. Unfortunately, investigating the underlying cause of this effect is beyond the scope of this work. Ultimately, it suggests that analysis of the *calibration* of autoregressive structured prediction models is an interesting area of future research.

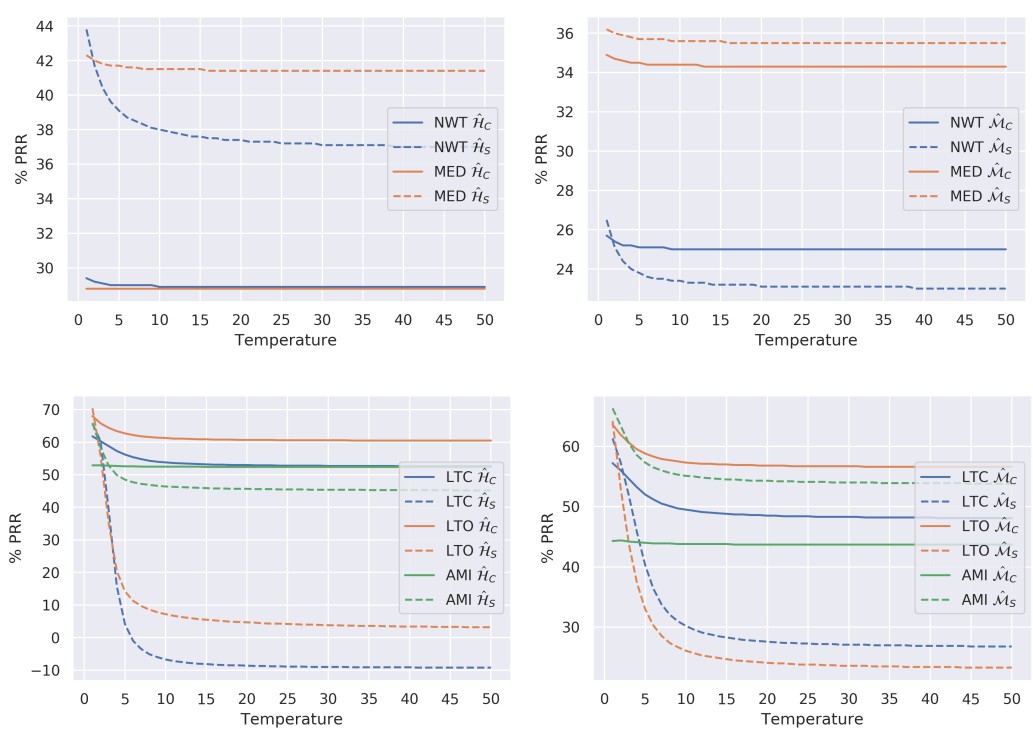

Figure 10: Sensitivity of uncertainty measures to importance weighting calibration.

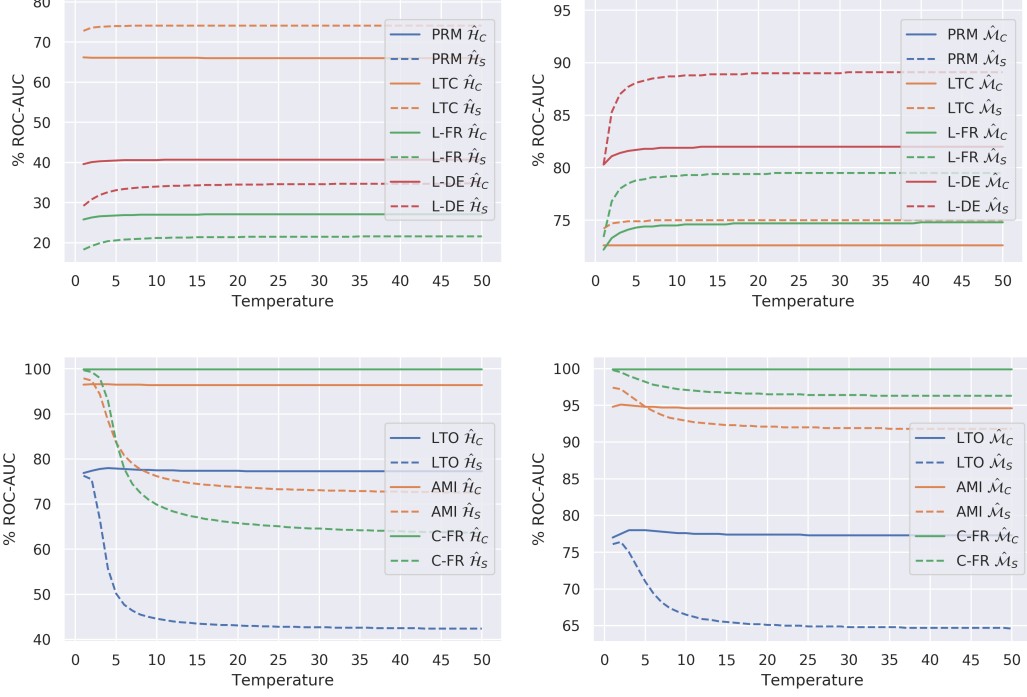

Figure 11: Sensitivity of uncertainty measures to importance weighting calibration.

## H  EFFECTS OF LENGTH-NORMALIZATION

In section 3 we make the claim that length-normalization in important - in this section we provide evidence in support of that claim. Here we compare sequence error detection and OOD detection performance for WMT'17 EN-DE NMT and LibriSpeech ASR systems when using the standard length-normalized versions of the uncertainties as well as their non-length-normalized counterparts. In these experiments we only consider translation/transcription hypotheses obtained via beam-search decoding from a *product-of-expectations* ensemble, and uncertainty measures obtained using the same combination approach.

Table 16: Effect of length-normalization on sequence-level Error Detection % PRR.

| Task | Test set | len-norm | ENS-PrEx TU $\hat{\mathcal{H}}_{\text{C-IW}}^{(1)}$ | $\hat{\mathcal{H}}_{\text{S-IW}}^{(1)}$ | ENS-PrEx KU $\hat{\mathcal{I}}_{\text{C-IW}}^{(1)}$ | $\hat{\mathcal{K}}_{\text{C-IW}}^{(1)}$ | $\hat{\mathcal{M}}_{\text{C-IW}}^{(B)}$ | $\hat{\mathcal{M}}_{\text{S-IW}}^{(1)}$ |
|---|---|---|---|---|---|---|---|---|
| LSP | LTC | - | 48.8 | 56.3 | 47.6 | 46.7 | 46.4 | 54.3 |
| | | + | 64.7 | **68.5** | 62.8 | 61.1 | 60.5 | 64.3 |
| | LTO | - | 49.6 | 54.4 | 48.2 | 46.6 | 46.0 | 53.0 |
| | | + | 73.5 | **76.0** | 72.3 | 69.5 | 68.6 | 70.5 |
| | AMI | - | 6.3 | 26.6 | -1.2 | -1.2 | -1.2 | 25.4 |
| | | + | 57.9 | **66.8** | 53.5 | 51.2 | 50.1 | 63.1 |
| ENDE | nwt14 | - | 20.2 | 30.7 | 24.2 | 23.9 | 23.6 | 27.8 |
| | | + | 29.9 | **46.6** | 29.1 | 28.1 | 27.4 | 30.4 |
| | MED | - | 21.0 | 33.6 | 28.6 | 28.5 | 28.3 | 32.7 |
| | | + | 31.1 | **45.5** | 37.1 | 36.4 | 35.8 | 38.9 |

The results in tables 16-17 definitively show, with one exception, that length-normalization consistently boosts the performance on all tasks. The only exception in for OOD detection when French (L-FR) input sentences are given to the ensemble. In this case there is a large improvement from *not* using length normalization. This seems odd, given than when German input sentences are using (in and En-De system) the same effect does not appear. Not that in both of these cases the pathological copy-through effect appears. However, it seems likely that the copy-through effect is far more pronounced for L-FR. The likely reason for length-norm proving detrimental is for long sentences - as the length of the translation increases, so does the chance of a token which is *not* copied successfully, on which the models will yield a far higher estimate of ensemble diversity. Length-normalization would mask such an effect relative.

Table 17: Effect of length-normalization on OOD Detection % ROC-AUC for ASR and NMT.

| Task | OOD Data | $T$ | B | Len-Norm | ENS-PrEx TU $\hat{\mathcal{H}}_{\text{C-IW}}^{(B)}$ | $\hat{\mathcal{H}}_{\text{S-IW}}^{(B)}$ | ENS-PrEx KU $\hat{\mathcal{I}}_{\text{C-IW}}^{(B)}$ | $\hat{\mathcal{K}}_{\text{C-IW}}^{(B)}$ | $\hat{\mathcal{M}}_{\text{C-IW}}^{(B)}$ | $\hat{\mathcal{M}}_{\text{S-IW}}^{(B)}$ |
|---|---|---|---|---|---|---|---|---|---|---|
| ASR | LTO | 1 | 20 | - | 73.2 | 73.5 | 73.4 | 73.9 | 74.0 | 74.3 |
| | | | | + | 76.9 | 76.3 | 76.6 | **77.0** | **77.0** | 76.1 |
| | AMI | 1 | 20 | - | 85.2 | 89.8 | 81.4 | 81.2 | 81.1 | 86.9 |
| | | | | + | 96.5 | **97.9** | 94.9 | 94.9 | 94.8 | 97.4 |
| | C-FR | 1 | 20 | - | 99.4 | 99.4 | 99.2 | 99.2 | 99.2 | 99.5 |
| | | | | + | **99.9** | 99.7 | **99.9** | **99.9** | **99.9** | 99.8 |
| NMT | LTC | 10 | 5 | - | 52.1 | 55.5 | 60.4 | 60.8 | 61.2 | 66.4 |
| | | | | + | 66.0 | 74.1 | 73.2 | 72.9 | 72.6 | **75.0** |
| | PRM | 10 | 5 | - | 65.6 | 68.7 | 86.8 | 88.3 | 89.1 | 92.1 |
| | | | | + | 82.9 | 82.7 | 96.7 | **96.9** | **96.9** | 96.5 |
| | L-FR | 10 | 5 | - | 63.7 | 50.9 | 78.4 | 81.3 | 83.0 | **87.7** |
| | | | | + | 27.1 | 21.7 | 65.2 | 71.2 | 74.8 | 79.6 |
| | L-DE | 10 | 5 | - | 44.9 | 37.1 | 70.1 | 73.8 | 76.1 | 85.8 |
| | | | | + | 40.8 | 34.9 | 76.1 | 79.9 | 82.1 | **89.2** |

## I    COMPARISON TO HEURISTIC MEASURES OF UNCERTAINTY

In this work we considered a range of information-theoretic measures of uncertainty, describe them theoretically in section 2 and provide Monte-Carlo estimators in section 3. However, in (Xiao et al., 2019; Fomicheva et al., 2020; Wang et al., 2019) a range of other measures was considered, though their properties were not analyzed. Firstly, (Xiao et al., 2019) considered computing the *cross-BLEU* or 'BLEU-Variance' of an ensemble. Here, the average square of the complement of pairwise sentence-BLEU between 1-best hypotheses $\hat{\boldsymbol{y}}^{(m)}$ produced by each individual model in the ensemble is used as a measure of uncertainty:

$$\text{X-BLEU} = \frac{1}{M(M-1)} \sum_{m=1}^{M} \sum_{q \neq m}^{M} \left(100 - \text{BLEU}(\hat{\boldsymbol{y}}^{(m)}, \hat{\boldsymbol{y}}^{(q)})\right)^2 \tag{44}$$

Similarly, an equivalent measure of ASR can be derived, which we will call *cross-WER*:

$$\text{X-WER} = \frac{1}{M(M-1)} \sum_{m=1}^{M} \sum_{q \neq m}^{M} \left(\text{WER}(\hat{\boldsymbol{y}}^{(m)}, \hat{\boldsymbol{y}}^{(q)})\right)^2 \tag{45}$$

These two measure of uncertainty assess the diversity between the 1-best hypotheses of each model in an ensemble. In this way they are conceptually related to measures of *knowledge uncertainty*. Notably, as they are only sensitive to the 1-best hypothesis, they are 'hard' versions of ensemble diversity, as they operate on the surface forms, rather than over probabilities. While a heuristically sensible measure, the main detraction is that this requires doing beam-search inference of *each* model in an ensemble, which is expensive and undesirable. Especially if the final prediction is derived through inference of the joint-ensemble.

Additionally, Fomicheva et al. (2020); Wang et al. (2019) consider the variance of the sentence-level probability and length-normalized log-probability of the 1-best hypothesis across models in an ensemble:

$$\mathbb{V}[P] = \mathbb{V}_{\mathsf{q}(\boldsymbol{\theta})}\left[\text{P}(\boldsymbol{y}_{\text{1-best}}|\boldsymbol{x}, \boldsymbol{\theta})\right] \tag{46}$$

$$\hat{\mathbb{V}}[\ln P] = \mathbb{V}_{\mathsf{q}(\boldsymbol{\theta})}\left[\frac{1}{L}\ln \text{P}(\boldsymbol{y}_{\text{1-best}}|\boldsymbol{x}, \boldsymbol{\theta})\right] \tag{47}$$

These are also measures of diversity, and therefore *knowledge uncertainty*. However, they are not strictly information-theoretically meaningful and it is not how to relate them to concepts, such as entropy. Given these two measures, it is unclear how to include information from other hypotheses and whether length-normalization is necessary. First we address the latter - we consider a length-normalized version of variance, and the non-length-normalized variance of the log-probability.

$$\hat{\mathbb{V}}[P] = \mathbb{V}_{\mathsf{q}(\boldsymbol{\theta})}\left[\text{P}(\boldsymbol{y}_{\text{1-best}}|\boldsymbol{x}, \boldsymbol{\theta})^{\frac{1}{L}}\right] \tag{48}$$

$$\mathbb{V}[\ln P] = \mathbb{V}_{\mathsf{q}(\boldsymbol{\theta})}\left[\ln \text{P}(\boldsymbol{y}_{\text{1-best}}|\boldsymbol{x}, \boldsymbol{\theta})\right] \tag{49}$$

$$\tag{50}$$

Secondly, we extend all four of these measures to average variances of the hypotheses within the beam as follows:

$$\mathbb{V}^{(B)}[P] = \sum_{b=1}^{B} \pi_b \mathbb{V}_{\mathsf{q}(\boldsymbol{\theta})}\left[\text{P}(\boldsymbol{y}^{(b)}|\boldsymbol{x}, \boldsymbol{\theta})\right], \, \boldsymbol{y}^{(b)} \in \mathcal{B} \tag{51}$$

$$\hat{\mathbb{V}}^{(B)}[P] = \sum_{b=1}^{B} \pi_b \mathbb{V}_{\mathsf{q}(\boldsymbol{\theta})}\left[\text{P}(\boldsymbol{y}^{(b)}|\boldsymbol{x}, \boldsymbol{\theta})^{\frac{1}{L^{(b)}}}\right], \boldsymbol{\theta})\right], \boldsymbol{y}^{(b)} \in \mathcal{B} \tag{52}$$

$$\mathbb{V}^{(B)}[\ln P] = \sum_{b=1}^{B} \pi_b \mathbb{V}_{\mathsf{q}(\boldsymbol{\theta})}\left[\ln \text{P}(\boldsymbol{y}^{(b)}|\boldsymbol{x}, \boldsymbol{\theta})\right], \boldsymbol{\theta})\right], \, \boldsymbol{y}^{(b)} \in \mathcal{B} \tag{53}$$

$$\hat{\mathbb{V}}^{(B)}[\ln P] = \sum_{b=1}^{B} \pi_b \mathbb{V}_{\mathsf{q}(\boldsymbol{\theta})}\left[\frac{1}{L^{(b)}}\ln \text{P}(\boldsymbol{y}^{(b)}|\boldsymbol{x}, \boldsymbol{\theta})\right], \boldsymbol{\theta})\right], \, \boldsymbol{y}^{(b)} \in \mathcal{B} \tag{54}$$

Thus, we explore the effect of considering additional hypotheses and whether these measures need length-normalization or not.

Table 18 explores the utility of these uncertainty measures and compares them to the best-performing measures of knowledge uncertainty uncertainty - reverse mutual information on the task of OOD input detection. The results show that, with one exception, estimates of reverse mutual information outperform these 'heuristic' measures. With regards to the measures - it is clear that length-normalization, with one exception, improves performance. We can also see that these heuristic measures can be improved by considering importance-weighted averages across the hypotheses within the beam. However, these gains are inconsistent, and sometimes this is detrimental. This highlights both the value of information theoretic measures, whose behaviour is far more consistent.

Table 18: Comparison of information-theoretic and heuristic measures on OOD detection (% ROC-AUC).

| Task | OOD Data | $T$ | $B$ | Info.Theor. | | Heuristic | | | | X-BLEU | X-WER |
|---|---|---|---|---|---|---|---|---|---|---|---|
| | | | | $\hat{\mathcal{M}}_{\text{C-IW}}^{(B)}$ | $\hat{\mathcal{M}}_{\text{S-IW}}^{(B)}$ | $\mathbb{V}^{(B)}[P]$ | $\hat{\mathbb{V}}^{(B)}[P]$ | $\mathbb{V}^{(B)}[\ln P]$ | $\hat{\mathbb{V}}^{(B)}[\ln P]$ | | |
| ASR | LTO | 1 | 1 | 76.6 | 73.9 | 52.6 | 72.0 | 71.0 | 72.7 | 74.3 | 71.8 |
| | | | 20 | **77.0** | 76.1 | 51.2 | 72.7 | 73.8 | 74.6 | | |
| | AMI | 1 | 1 | 96.2 | 96.4 | 41.5 | 87.3 | 86.1 | 95.8 | 95.9 | 95.8 |
| | | | 20 | 94.8 | **97.4** | 42.3 | 85.6 | 84.0 | 96.7 | | |
| | C-FR | 1 | 1 | **99.9** | 99.8 | 7.9 | 82.0 | 97.6 | 98.0 | 99.5 | 99.7 |
| | | | 20 | **99.9** | 99.8 | 12.5 | 77.6 | 99.0 | 98.4 | | |
| NMT | LTC | 10 | 1 | 72.2 | 73.3 | 46.0 | 68.7 | 65.6 | 72.3 | 65.1 | 58.2 |
| | | | 5 | 72.6 | **75.0** | 46.1 | 68.6 | 66.5 | 72.5 | | |
| | PRM | 10 | 1 | 96.7 | 96.2 | 32.7 | 88.8 | 90.4 | 93.4 | 80.8 | 73.5 |
| | | | 5 | **96.9** | 96.5 | 33.7 | 88.7 | 91.5 | 93.0 | | |
| | L-FR | 10 | 1 | 72.1 | 70.9 | 59.2 | 76.4 | 85.1 | 71.1 | 46.8 | 52.7 |
| | | | 5 | 74.8 | **79.6** | 58.0 | 77.7 | 89.4 | 72.2 | | |
| | L-DE | 10 | 1 | 80.1 | 76.0 | 69.1 | 82.7 | 76.9 | 78.3 | 36.1 | 38.7 |
| | | | 5 | 82.1 | 89.2 | 69.8 | **90.0** | 86.0 | 86.2 | | |

## J  CHECKPOINT ENSEMBLES

This work focused mainly on ensemble of models constructed by training from different random initializations. While this tends to yield the best ensembles, this is an extremely expensive process, especially for large transformer models trained on industrial scale corpora. Thus, in this section we conduct a preliminary investigation of *checkpoint ensembles* - ensembles of models constructed by considering different checkpoints within a *single* training run. In this work we consider the checkpoints for the 10 last epochs for NMT model training. For ASR, we *checkpoint average* 5 checkpoints at 5-checkpoint intervals within the last 30 epochs, yielding an ensemble of 6 checkpoints, where each is an average of five checkpoints. Results in tables 19-20 shows that checkpoint ensemble yield better predictive performance than *single models* (CPT-AVG), on average. However, random-init ensembles (those considered in the main work), yield consistently superior performance.

Table 19: Predictive performance in terms of BLEU, %WER and NLL on newstest14 and LibriSpeech.

| Model | NWT'14 BLEU | | MED BLEU | | NWT'14 NLL | | MED NLL | |
|---|---|---|---|---|---|---|---|---|
| | EN-DE | EN-FR | EN-DE | EN-FR | EN-DE | EN-FR | EN-DE | EN-FR |
| CPT-AVG | $28.8_{\pm0.2}$ | $45.4_{\pm0.3}$ | $29.9_{\pm0.5}$ | $51.3_{\pm0.5}$ | $1.46_{\pm0.02}$ | $1.10_{\pm0.01}$ | $1.29_{\pm0.01}$ | $0.82_{\pm0.01}$ |
| CPT-ENS | $29.3_{\pm0.1}$ | - | $30.5_{\pm0.3}$ | - | $1.42_{\pm0.01}$ | - | $1.25_{\pm0.00}$ | - |
| RND-ENS | **30.1** | **46.5** | **32.1** | 52.7 | **1.33** | **1.04** | **1.16** | **0.77** |

We also compare checkpoint ensembles with random-init ensembles of OOD detection. The results, which are a pleasant surprise, show that while checkpoint ensembles are consistently inferior, they are only marginally so. The biggest differences occur on NMT L-FR and L-DE datasets, where the pathological 'copy-through' effect kicks in. However, even here the difference between the best CPT-ENS and RND-ENS measures is only about 6% ROC-AUC points. This shows that the approach considered in this work need to be too expensive for practical application, and useful ensembles can

Table 20: Predictive performance in terms of BLEU, %WER and NLL on newstest14 and LibriSpeech.

| Model | ASR % WER | | | ASR NLL | | |
|---|---|---|---|---|---|---|
| | LTC | LTO | AMI | LTC | LTO | AMI |
| CPT-AVG | 5.6 ±0.2 | 14.7 ±0.5 | 78.7 ±13.4 | 0.34 ±0.00 | 0.86 ±0.02 | 5.78 ±0.32 |
| CPT-ENS | 5.3 ± NA | 13.9 ± NA | 54.0 ± NA | 0.27 ± NA | 0.67 ± NA | 4.68 ± NA |
| RND-ENS | **4.2** | **11.3** | **50.4** | **0.20** | **0.48** | **4.05** |

be formed even from the last 10 checkpoints of a *standard transformer model training run*, enabling usage in extreme-scale industrial applications.

Table 21: OOD Detection % ROC-AUC in Beam-Search decoding regime for ASR and NMT.

| Task | Test set | ENSM type | ENS-PrEx TU | | ENS-ExPr TU | | ENS-PrEx KU | | | | ENS-ExPr KU | | | |
|---|---|---|---|---|---|---|---|---|---|---|---|---|---|---|
| | | | $\hat{\mathcal{H}}_{\text{C-IW}}^{(B)}$ | $\hat{\mathcal{H}}_{\text{S-IW}}^{(B)}$ | $\hat{\mathcal{H}}_{\text{C-IW}}^{(B)}$ | $\hat{\mathcal{H}}_{\text{S-IW}}^{(B)}$ | $\hat{\mathcal{I}}_{\text{C-IW}}^{(B)}$ | $\hat{\mathcal{H}}_{\text{C-IW}}^{(B)}$ | $\hat{\mathcal{M}}_{\text{C-IW}}^{(B)}$ | $\hat{\mathcal{M}}_{\text{S-IW}}^{(B)}$ | $\hat{\mathcal{I}}_{\text{C-IW}}^{(B)}$ | $\hat{\mathcal{H}}_{\text{C-IW}}^{(B)}$ | $\hat{\mathcal{M}}_{\text{C-IW}}^{(B)}$ | $\hat{\mathcal{M}}_{\text{S-IW}}^{(B)}$ |
| ASR | LTO | CPT-ENS | 75.1 | 74.7 | 75.0 | 74.9 | 74.6 | 74.6 | 74.5 | 74.0 | 74.1 | 74.6 | 74.6 | 73.7 |
| | | RND-ENS | 76.9 | 76.3 | 76.4 | 77.0 | 76.6 | 77.0 | 77.0 | 76.1 | 74.8 | 76.7 | 76.9 | 75.3 |
| | AMI | CPT-ENS | 95.6 | 96.0 | 95.6 | 96.0 | 93.3 | 92.8 | 92.6 | 94.1 | 92.8 | 92.8 | 92.5 | 93.9 |
| | | RND-ENS | 96.5 | 97.9 | 96.4 | 97.9 | 94.9 | 94.9 | 94.8 | 97.4 | 93.0 | 94.8 | 94.8 | 97.0 |
| | C-FR | CPT-ENS | 99.9 | 99.5 | 99.9 | 99.6 | 99.8 | 99.7 | 99.7 | 99.2 | 97.9 | 99.7 | 99.7 | 98.5 |
| | | RND-ENS | 99.9 | 99.7 | 99.9 | 99.8 | 99.9 | 99.9 | 99.9 | 99.8 | 89.7 | 99.9 | 99.9 | 99.1 |
| ENDE | LTC | CPT-ENS | 63.7 | 72.5 | 63.5 | 72.4 | 70.9 | 70.7 | 70.6 | 75.0 | 59.9 | 67.0 | 70.5 | 73.0 |
| | | RND-ENS | 66.0 | 74.1 | 65.0 | 74.0 | 73.2 | 72.9 | 72.6 | 75.0 | 58.1 | 69.3 | 72.1 | 73.9 |
| | PRM | CPT-ENS | 81.1 | 81.3 | 80.8 | 81.5 | 96.4 | 96.4 | 96.3 | 94.7 | 79.0 | 93.4 | 96.2 | 92.9 |
| | | RND-ENS | 82.9 | 82.7 | 80.7 | 84.5 | 96.7 | 96.9 | 96.9 | 96.5 | 72.1 | 95.0 | 96.7 | 95.2 |
| | L-FR | CPT-ENS | 29.5 | 27.8 | 28.9 | 27.7 | 60.4 | 62.9 | 64.6 | 73.2 | 33.3 | 47.2 | 65.0 | 74.2 |
| | | RND-ENS | 27.1 | 21.7 | 23.0 | 22.9 | 65.2 | 71.2 | 74.8 | 79.6 | 23.4 | 50.1 | 73.8 | 80.7 |
| | L-DE | CPT-ENS | 39.4 | 35.9 | 38.5 | 35.7 | 72.3 | 73.3 | 74.0 | 82.3 | 45.9 | 61.5 | 73.9 | 82.0 |
| | | RND-ENS | 40.8 | 34.9 | 36.1 | 38.3 | 76.1 | 79.9 | 82.1 | 89.2 | 41.9 | 73.3 | 81.2 | 88.8 |

