# OpenReview forum: "Uncertainty Estimation in Autoregressive Structured Prediction"
_ICLR.cc/2021/Conference — ICLR 2021 Poster_

### Official Review · AnonReviewer1 · 2020-10-27
**New measure of epistemic uncertainty quantification for structured prediction but novelties are incremental**

**Rating:** 6
**Confidence:** 4

**Review:**

This paper proposes two different measures of knowledge (epistemic) uncertainty in structured prediction with an autoregressive model and discusses how to compute their approximations. The main contribution is the proposed reverse mutual information (RMI) as a measure of epistemic uncertainty in structured prediction. Experiments on benchmark datasets demonstrate the effectiveness of the proposed method in error detection and OOD detection.


### Clarity
#### Pros
- This paper clearly states the main contributions, assumptions, experimental settings.

#### Cons
- The naming of reverse mutual information is confusing since mutual information itself is symmetric, unlike KL-divergence.
- Table 1 shows the predictive performance with an ensemble of autoregressive models, however, it is unrelated to the main contribution of the paper. The proposed RMI is not used to improve the performance of the ensemble of autoregressive models.
- Eq(4), $\tilde{\theta}$ is used without definition. And there is no $\mathcal{D}$ on the right side of the equation. Why is $q(\theta) = q(\tilde{\theta})$?
- In Eq(12), how is $y$ sampled from $P(y\mid x, \mathcal{D})$? Do you mean to sample from $q(\theta)$ instead of $p(\theta\mid \mathcal{D})$?
- Though the two ways, EP and PE, for combination of the ensemble of autoregressive models are both reasonable, only EP can be derived from assumptions in Eq (7). PE assumes the probability $P_{PE}(\boldsymbol{y} \mid \boldsymbol{x}, \mathcal{D})$ factorizes over the conditional probability ${P}(y_{l} \mid \boldsymbol{y}_{<l}, \boldsymbol{x}, \mathcal{D})$.
This should be made clear, otherwise can be confusing.

- On page 4, above Eq (10), it is claimed that
> (8) only considers the probabilities of individual tokens $y_{l}^{(s)}$ along a hypothesis $y^{(s)}$ while (9) considers the entire conditional distribution over each $y_{l}$.

  It is not clear why this is the case. In my understanding, (8) computes the entropy of the joint distribution $P(\boldsymbol{y} \mid \boldsymbol{x}, \mathcal{D})$,
which considers all the conditional distribution $\mathrm{P}\left(y_{l} \mid \boldsymbol{y}_{<l}, \boldsymbol{x}\right)$, no matter using EP or PE. Correct me if I am wrong.

- (Minor)
Typos:
  - Page 4,
above Eq (15),
practical considerations
as it yields poor predictive performance ? -> remove ?
  - Page 4,
  above Eq (10), while (8) yields A -> while (8) yields a
  - Page 4,
  below Eq (13), like (8)(9) -> like (8)(10)
  - Page 5,
above Eq (16),
the models can combined -> the models can be combined


### Originality
#### Pros
- The paper makes clear its main contributions, including introductions of different measures of uncertainty in structured prediction, the examination of two choices of the combination of ensemble models. The proposed reverse mutual information (RMI) improves the performance in OOD detection in structured prediction.

#### Cons
- The contribution of the paper is minor with the proposed RMI measure.
- The proposed RMI for epistemic uncertainty estimation is not used to improve the performance of the ensemble of autoregressive models.


### Significance
#### Pros
- Experimental results show improvement of the proposed method over baseline methods.

#### Cons
- The paper presents ablation studies of different methods for knowledge uncertainty estimation, but it is unclear whether the proposed method achieves state-of-the-art performance on different datasets for error detection and OOD detection. For example, compare to MC-dropout and other recently proposed methods without structured prediction with autoregressive models.
- The contribution is not very significant, but mostly an extension and application of uncertainty measures to autoregressive models.
For example, the reverse mutual information (RMI), seems to be related to the Reverse KL-divergence proposed in [1]. More discussion is suggested on how the reverse version improves over the original version as is done in [1].
- Besides OOD detection, it is also interesting to show whether the proposed uncertainty measure can improve other tasks, such as adversarial sample detection, and active learning. For example, [2] presents more thorough experiments in these different tasks.
- The paper proposes two ways to combine the ensemble of autoregressive models, EP or PE. Experimental results are provided on these two ways of combination.
However, it would be better if more theoretical analysis to compare these two could be given based on different assumptions. For example, analysis of the computational complexity of the two choices.
- Similarly, it would be better if complexity analysis can be provided for computing approximations of the entropy using relative-entropy chain-rule in Eq (9),(11),(13), and using the traditional ways in Eq (8),(10),(12). Since it is claimed at "no extra cost" in the paper.
- It is discussed in the paper that EPKL, RMI, MI all measures knowledge uncertainty, and RMI = EPKL - MI. And only MI 'cleanly' decomposes into total and data uncertainty.
A natural question is that if we decompose total uncertainty using RMI, i.e. Total uncertainty = data uncertainty + RMI + res, what does the remaining term res represent?
For example, if RMI gives a better measure of knowledge uncertainty as shown empirically in the paper, then the remaining term in the total uncertainty may measure other types of uncertainty rather than data uncertainty or knowledge uncertainty? Or there are other ways to represent total uncertainty and/or data uncertainty correspondingly?
Correct me if I am wrong.
And it would be better if more discussion can be provided in the paper.

[1] Reverse kl-divergence training of prior networks: Improved uncertainty and adversarial robustness

[2] SDE-Net: Equipping Deep Neural Networks with Uncertainty Estimates

---

> ### Author Response · Authors · 2020-11-13
> **Response to Reviewer 1 - Part 1.**
>
> Dear Reviewer1, we are grateful for your hard work and detailed review. However, we feel that we have failed to properly convey the significance of our contribution to you. Let us attempt to resolve this misunderstanding.
>
> While ensemble approaches and information-theoretical measures of uncertainty have been examined in the past, *no prior work* has ever rigorously examined ensemble based uncertainty estimation in the context of structured prediction tasks. The key contribution of our work is “the examination of the attributes of ensemble-based uncertainty estimation for autoregressive structured prediction tasks”, which is novel because it has never been rigorously studied before, especially not for ASR. Furthermore, this work represents the largest-scale application of Bayesian-inspired uncertainty estimation, which in itself is significant. This is the key novelty and contribution of our work. Though we feel that it is an *important* contribution, the proposal of a new uncertainty measure (RMI) is NOT our *main* contribution.
>
> We will also agree each of your comments point-by-point.
>
> CLARITY
>
> 1. We chose the name Reverse Mutual Information because we have flipped the direction of the KL-divergence (which is non-symmetric), relative to standard MI.
>
> 2. While predictive performance is not the main contribution of the paper, it is always necessary to quote predictive performance in order to demonstrate that the models under consideration are sensible. In addition, we use this to explore the impact of how an ensemble is combined.
>
> 3. This is a notational convenience. This is supposed to indicate a different dummy variable, as we are comparing pairwise KL divergence between samples from the same distribution over models (from the same ensemble). D is absorbed into q(\theta)
>
> 4. We sample translation/transcription hypotheses from P(y∣x,D) via beam-search decoding. Beam-search decoding can be interpreted as a form of importance sampling.
>
> 5. We will address the confusion regarding eq.7 and PE vs EP in the text.
>
> 8. Equations 8 and 9 represent Monte-Carlo estimators of entropy which yield the same values *in the limit as S -> infinity*. However, when we consider a finite number of hypotheses, 8 only considers the probabilities of tokens which occur along the hypothesis. On the other hand, 9 estimates the conditional entropy over the full conditional distribution over the next token (over the entire vocabulary) at each step along the hypothesis. Thus, 8 consider LS values, while 9 consider LSK values (K is the size of vocabulary).
>
> 9. Typos - Many thanks for finding these typos! Already fixed.
>
> ORIGINALITY
>
> 1. As stated above, the main contribution is not RMI, but the examination of general ensemble-based uncertainty estimation in the context of structured prediction.
>
> 2. The goal of uncertainty estimation is orthogonal to improving the predictive performance of a system. RMI is a measure of knowledge uncertainty for which an asymptotically exact Monte-Carlo estimator can be obtained, and yields the most competitive OOD detection performance.
>
> SIGNIFICANCE - responses in second part.

---

> > ### Author Response · Authors · 2020-11-13
> > **Response to Reviewer 1 - Part 2**
> >
> > SIGNIFICANCE
> >
> > 1. Again, just to be clear - the focus of this work is exclusively ensemble-based uncertainty estimation in structured prediction models, specifically autoregressive models. We explore the different attributes of this approach, derivable uncertainty measures and so on. Critically, *no other work* has considered the structured uncertainty estimation tasks examined in this paper. Thus, while we dislike the term, our work defines the de-facto state of the art.
> >
> > MC-Dropout [6] is an *ensemble generation method*, which has been repeatedly shown to be inferior to Deep-Ensembles [11], which are the current go-to SOTA ensemble-generation scheme [7,8,9]. Deep-Ensembles is the ensemble-generation scheme we use in this work.
> >
> > 2. RMI is unrelated to the Reverse KL divergence (RKL) loss in [1]. RMI is an uncertainty measure for ensembles, while the RKL loss is a training criterion for Dirichlet Prior Networks.
> >
> > 3. We very much agree that the application of uncertainty estimates to adversarial attack detection and active learning is meaningful and interesting. This has received attention in non-structured prediction tasks in images [1,2,10].  However, to our knowledge, adversarial attacks are challenging to define and implement for structured sequence-based tasks, such as ASR and NMT,  and deserve a thorough and meaningful investigation all on their own. The same goes for active learning - Bayesian approaches to active learning have mostly focused on very simple, small-scale image datasets such as MNIST [3,4]. Their generalisation to large-scale structured problems such as NMT and ASR deserves an an extensive investigation all on their own.
> >
> > Thus, while uncertainty for detection of adversarial examples and active learning in structured prediction are very interesting, both are beyond the scope of the current investigation.
> >
> > 4. We agree that it would be great to have a theoretical analysis of the difference between these two ensemble combinations. However, despite months of trying, we have yet to find any way to relate the two theoretically. There are no bounds or inequalities which can be used to relate the two. The difference seems to be entirely empirical - a consequence of the choice of model, the training method, etc… Alternatively, this may be a consequence of the locally normalised nature of autoregressive models or the fact the model’s representation of context is imperfect, as noted in our response to reviewer 3.
> >
> > 5. All measures of uncertainty are obtainable at identical cost. In fact, all information necessary to calculate these uncertainties is available (and usually discarded) during beam search decoding of an ensemble, hence “no extra cost” - we do not need to take extra steps to evaluate these uncertainties, we simply need to keep track of the quantities of interest during decoding. Though there is negligible overhead for keeping track of the quantities which are usually discarded.
> >
> > 6. All uncertainty in machine learning models is a result of either data (aleatoric) uncertainty and knowledge (epistemic) uncertainty [5,6]. There are no other sources of uncertainty. Playing with the above decomposition simply “shuffles around” data and knowledge uncertainty.
> >
> > To answer your question “ i.e. Total uncertainty = data uncertainty + RMI + res, what does the remaining term res represent?” - it is EPKL.
> >
> > We believe you have made a sign error (+RMI vs -RMI). From the relationship above:
> >
> > MI =  TU - DU = EPKL - RMI
> > TU = DU - RMI + RES.
> > RES = EPKL.
> >
> >
> >
> > [1] Reverse kl-divergence training of prior networks: Improved uncertainty and adversarial robustness
> >
> > [2] SDE-Net: Equipping Deep Neural Networks with Uncertainty Estimates
> >
> > [3] Deep Bayesian Active Learning with Image Data. Gal et al.
> >
> > [4] BatchBALD: Efficient and Diverse Batch Acquisition for Deep Bayesian Active Learning. Kirsch et al.
> >
> > [5] Uncertainty Estimation in Deep Learning with Application to Spoken Language Assessment. A Malinin PhD Thesis.
> >
> > [6] Uncertainty in Deep Learning. Y. Gal PhD Thesis.
> >
> > [7] Can you trust your model’s uncertainty? Evaluating predictive uncertainty under dataset shift.
> >
> > [8] Pitfalls of in-domain uncertainty estimation and ensembling in deep learning.
> >
> > [9] Deep ensembles: A loss landscape perspective.
> >
> > [10] Understanding Measures of Uncertainty for Adversarial Example Detection
> >
> > [11] Simple and Scalable Predictive UncertaintyEstimation using Deep Ensembles

---

> > > ### Comment · AnonReviewer1 · 2020-11-18
> > > **updated review**
> > >
> > >
> > > ### Updated review
> > > I appreciate the authors' hard work. However, below are some questions/suggestions I still have.
> > >
> > > #### Major
> > > 1. Total uncertainty decomposition. (Though this question is only for discussion purposes and may not affect very much the final score of the paper, I feel it is important to clarify.)
> > >
> > > I understand that MI = -RMI+EPKL and we can directly plug in to get Total uncertainty = data (aleatoric) uncertainty -RMI + EPKL. It also makes sense that RMI and EPKL can measure epistemic uncertainty based on their definitions. However, playing with this equation in this way is not the intention of my question.
> > > The question is I do not understand the decomposition Total uncertainty = aleatoric uncertainty -RMI + EPKL. If RMI$\geq 0$ and we use RMI to better represents epistemic uncertainty, the larger RMI the larger the epistemic uncertainty, then we can replace MI with RMI (no minus sign) in the equation to get Total uncertainty = aleatoric uncertainty + RMI + res. I am asking for an explainable practical meaning for the decomposition of uncertainty.
> > >
> > > Following the claim that there are only two types of uncertainty, then Total uncertainty = aleatoric uncertainty + epistemic uncertainty. And if RMI better measures epistemic uncertainty, then plug RMI into the equation we can get Total uncertainty = aleatoric uncertainty + RMI + res. Then we need an explanation for res. Otherwise why RMI is a better measure of epistemic uncertainty if it cannot satisfy the requirement Total uncertainty = aleatoric uncertainty + epistemic uncertainty.
> > > Or I can also accept a decomposition such as Total uncertainty = data uncertainty +$k\cdot$RMI, $k>0$, but in such case, $k$ will be dependent on the input. Or we may need a better corresponding measure for total uncertainty and aleatoric uncertainty.
> > >
> > > 2. Whether there are other types of uncertainty. (Again this question is only for discussion purposes and may not affect very much the final score of the paper, but I feel it is important to clarify.)
> > >
> > > I also agree that there are two major types of uncertainty: aleatoric uncertainty and epistemic uncertainty. And Total uncertainty = Aleatoric uncertainty + Epistemic uncertainty.
> > > However, I believe previous and current works have also proposed different types or _subtypes_ of uncertainty.
> > > For example, in the paper [1], it derives Total uncertainty = data uncertainty (aleatoric uncertainty) + distributional uncertainty + model uncertainty (epistemic uncertainty). How do you understand this decomposition and distributional uncertainty?
> > > Another example is in the paper [2], which also proposes different categorizations of uncertainty.
> > >
> > > [1] Predictive Uncertainty Estimation via Prior Networks
> > >
> > > [2] Towards Principled Uncertainty Estimation for Deep Neural Networks
> > >
> > > #### Minor
> > >
> > > 1. Comparison to other methods for OOD detection.
> > >
> > > MC-dropout is only intended for an example. There are more recent methods that outperform deep ensembles such as [3].
> > > However, the intention of this question is mainly to ask for a comparison of OOD detection in the same dataset setting but _without structured prediction_ as a baseline using commonly used baselines such as deep ensembles and MC-dropout to make the paper more convincing. Because though previous studies [4,5,6] show improved performance of deep ensembles over MC-dropout, the studies are not on the same tasks presented in this paper. They mainly focus on computer vision related tasks.
> > > Again this is mainly a _suggestion_ to improve the paper.
> > >
> > > [3] Bayesian Deep Ensembles via the Neural Tangent Kernel
> > >
> > > [4] Simple and Scalable Predictive Uncertainty Estimation using Deep Ensembles
> > >
> > > [5] Pitfalls of in-domain uncertainty estimation and ensembling in deep learning
> > >
> > > [6] Deep ensembles: A loss landscape perspective

---

> > > > ### Author Response · Authors · 2020-11-20
> > > > **Response to R1's additional comments.**
> > > >
> > > > Regarding Major Comments
> > > >
> > > > 1. Ok, we see where you are coming from. Before we can answer the question about decomposition, we first need to clarify a possible misunderstanding. As defined in equations 3-5, we cannot say whether MI, EPKL or RMI are an a-prior ‘better” measure of epistemic uncertainty in any theoretical sense. Due to the non-symmetry of KL-divergence each asks an information theoretically different question and describe a different aspect of ensemble diversity:
> > > >
> > > > MI asks  “How well, on average, does the ensemble-mean represent each model in the ensemble?”.
> > > > RMI asks “How well, on average, does each model in the ensemble represent the ensemble mean?”
> > > > EPKL asks “How well, on average, does each model represent all other models in the ensemble?”
> > > >
> > > > In this sense, EPKL is the  “most complete” description of the mismatch, and therefore largest in absolute value. This also intuitively explains why EPKL = RMI + MI.
> > > >
> > > > RMI is only ‘better”  in a *practical* sense in the context of structured prediction. Specifically, we cannot use equations 3-5 directly in structured prediction - the hypothesis space is too vast, and we need to obtain Monte-Carlo estimates for each measure. For structured models obtaining hypotheses is a non-trivial, expensive, inference task. Thus, we only want to run inference (effectively, sample from) the joint ensemble. Given this restriction, we can obtain an asymptotically exact Monte-Carlo estimate *only* for RMI. In contrast, for EPKL and MI we obtain useful, but asymptotically incorrect approximations.
> > > >
> > > > Regarding decomposition - it is nice that our three measures of epistemic uncertainty tie up with each other, allowing us to say that we can obtain an information-theoretically complete description of the mismatch. However, we do not believe one should attempt to impose a requirement that TU = DU + KU for all measures of uncertainty. It is unclear what practical benefit can be derived from attempting to satisfy this requirement, as none of the measures are a-prior *better* in any theoretical sense.
> > > >
> > > > 2. As you say, this is indeed an interesting area for discussion, possibly beyond the scope of this rebuttal. Everything is a subtype of aleatoric or epistemic uncertainty, and different papers have sliced-and-diced this in different ways. The appropriate split is very dependent on the context of the problem being solved.
> > > >
> > > > With regards to [1]. Distributional uncertainty is a subtype of epistemic uncertainty. In more detail - epistemic occurs in two major situations. First, when the test-input comes from an altogether different distribution than from which the training data came from (dataset had cat and dogs, got a tractor as test input). Second, when the test-input comes from an “in-domain” region which was sparsely covered by training data. From a Bayesian point of view, these are degrees of the same thing. Distributional uncertainty as defined in [1] focuses specifically on the first situation. This is because Prior Networks treat all in-domain data as equally in-domain (they are not sensitive to data sparsity), and are given a specific “OOD-training set” which instructs the Prior Network where the boundary of the in-domain region is. In terms of the Prior Network paper, “model uncertainty” would cover the data sparsity situation.
> > > >
> > > > With regards to [2], they define “model uncertainty”, ‘intrinsic uncertainty” and “open set uncertainty. The latter two directly map to “data/aleatoric uncertainty” and “knowledge/epistemic uncertainty (distribution mismatch situation”. The other can likely be attributed to epistemic uncertainty.
> > > >
> > > > [1] Predictive Uncertainty Estimation via Prior Networks
> > > > [2] Towards Principled Uncertainty Estimation for Deep Neural Networks
> > > >
> > > >
> > > > Regarding Minor Comments:
> > > >
> > > > We are a little confused by this request and are unsure what you are suggesting. We cannot evaluate ensemble methods on the considered NMT/ASR datasets “without structured prediction” because NMT and ASR are *fundamentally* structured tasks, as you are transducing between two variance-length sequences. Perhaps you are asking for something else?

---

> > > > > ### Comment · AnonReviewer1 · 2020-11-21
> > > > > **updated discussion**
> > > > >
> > > > > Thank you for your discussion.
> > > > >
> > > > > ### Major
> > > > > 1. I totally agree that EPKL and RMI can measure epistemic uncertainty as stated in my previous comments. And in fact, there can be many different measures of epistemic uncertainty, eg. using variance [1], or replacing the divergence measures in MI. But satisfying TU = DU + KU is desirable as it is convenient for deriving subtypes of uncertainty and still keeps the decomposition "clean". For example, in [2,3], the subtypes of epistemic uncertainty is measured by mutual information (MI) and still keeps the decomposition clean. But this is not the case if replacing MI with RMI.  I understand RMI is ‘better' in a practical sense in the context of structured prediction. But in a more general sense, I believe satisfying TU = DU + KU is desirable (but may not be necessary).
> > > > >
> > > > > 2. No additional comments for this discussion. I think the authors' claims make sense though I do not completely agree with a clear separation between in-domain data sparsity and out-of-domain. Because there is ambiguity in the definition of out-of-domain data. For example, some hand written letters like "o" look very similar to hand-written digits like "0".
> > > > >
> > > > > ### Minor
> > > > > 1. Some intrinsically structured tasks can be handled in an unstructured manner. For example, in semantic segmentation [4], the ablation study is performed for unstructured model (FCN) assuming the output segmentation predictions are independent for each pixel. In your case, it is assuming the outputs $y_l$ are independent and directly learning $p(y_l | x)$. But I am not sure whether this is possible for your tasks. Again this is only a minor suggestion if it is possible.
> > > > >
> > > > > [1] What Uncertainties Do We Need in Bayesian Deep Learning for Computer Vision?
> > > > >
> > > > > [2] Predictive Uncertainty Estimation via Prior Networks
> > > > >
> > > > > [3] Accurate Uncertainty Estimation and Decomposition in Ensemble Learning
> > > > >
> > > > > [4] Conditional Random Fields as Recurrent Neural Networks

---

> > > > > > ### Author Response · Authors · 2020-11-23
> > > > > > **We seem to have reached a consensus. :)**
> > > > > >
> > > > > > Major
> > > > > >
> > > > > > 1. Oh, we definitely agree that TU = DU + KU is a conceptually desirable (and pleasing) property (for example the law of total variation does this in regression applications, such as [1]) which, unfortunately, is not always satisfiable, and may not be necessary in practice.
> > > > > >
> > > > > > 2. We only separated data-sparsity and OOD for diagrammatic purposes - we definitely don’t claim that there is a clear separation between data-sparsity and OOD, it’s more of a spectrum. We also completely agree that there are additional complexities, such as the OOD distribution not being fully disjoint with the in-domain distribution.
> > > > > >
> > > > > > Minor
> > > > > >
> > > > > > The issue is that in images the output space has a known, fixed, dimensionality (the size of the image). In NMT/ASR tasks the target sequence has variable dimensionality, and thus we cannot discard the structural component (though it can have a non-autoregressive structure, which would need an altogether different decoding mechanism).

---

> > ### Comment · AnonReviewer1 · 2020-11-18
> > **updated review**
> >
> > ### Updated review
> >
> > The authors have addressed some of my concerns. I appreciate the efforts. However, below are some concerns/suggestions I still have.
> >
> > #### Major
> >
> > 1. The main contribution of the paper.
> > The authors in the rebuttal claim the main contribution is “the examination of the attributes of ensemble-based uncertainty estimation for autoregressive structured prediction tasks”. I totally agree that this is an important problem. However,
> > first, I agree with Reviewer3 that the title "uncertainty in structured prediction" is a bit misleading. Because in this paper, the structured model is an autoregressive model, the task is speech-related. There are many more structured models such as Markov random field, graph neural networks, etc., and possible applications to many different tasks (eg. computer vision tasks). Therefore, claiming to be the "_largest-scale_ application of Bayesian-inspired uncertainty estimation" seems to be inappropriate.
> > Second, the 'examination' to me is more like an extension of the deep ensemble method and existing measures or approximations of estimating epistemic uncertainty and their empirical evaluation with a certain type of structured model for certain tasks rather than a big innovation.
> >
> > #### Minor
> > 1. The naming of reverse mutual information.
> > The name Reverse Mutual Information makes people ask why do we need Reverse MI since MI is symmetric. A better option for naming might be Expected Reverse KL divergence. But this is only a suggestion. The authors can choose what they think is better.
> >
> > 2. About the confusion in the equation (4).
> > I understand the meaning of Eq.(4) after the authors' explanation.
> > But the expression in this equation is still confusing. For example, the left side has no $\tilde{\theta}$ but has $\mathcal{D}$, and the right side of the equation has $\tilde{\theta}$ but no $\mathcal{D}$. I suggest the authors address this in the revision.

---

> > > ### Author Response · Authors · 2020-11-20
> > > **Response to Reviewer 1's additional comments.**
> > >
> > > Thank you for your response!
> > >
> > > Before we address the broader scale concerns you raise, we would first like to clarify that the methods and mathematics for uncertainty estimation in structured prediction proposed in our work can be used together with *any* ensemble-generation method, not just Deep Ensembles. We simply chose deep ensembles because they were simple, yield very competitive results and are overall convenient to work with. However, we could equivalently consider dropout, checkpoint ensembles (see appendix J), BatchEnsemble, etc… So our examination is more general than a simple extension of Deep Ensembles.
> > >
> > > Regarding your broader concerns. The choice of name has clearly raised more controversy and discussion than we wanted it to. Our motivation for choosing it was that autoregressive models represent a structured prediction model-type which is widely used and makes fairly weak conditional independence assumptions relative to models like HMMs, CTC, Markov Random fields, and so on, in the sense that they do not make Markovian assumptions (or equivalent make an Infinite-order Markov assumption). Given that we seem to be able to change the title of the work (to our surprise!), and if all the reviewers allow this, we are happy to moderate our title down to “Uncertainty Estimation in Autoregressive Structured Prediction”.
> > >
> > > We would like to clarify our statement regarding the question of the scale of our application. Our evaluation is done using large models on large-scale NMT and ASR tasks. Indeed we are unaware of any work on a larger scale task, where Bayesian methods have been applied to uncertainty estimation. In more detail:
> > >
> > > Firstly, Most work on Bayesian(-inspired) methods for uncertainty estimation has focused on small-scale tasks. Typically these are synthetic tasks, UCI regression datasets, MNIST (and similar), SVHN, CIFAR10/100. Very few works go to the level of ImageNet-1K. All of these tasks (with the exception of ImageNet), are far removed from typical production-scale scenarios, which feature more complex data, greater numbers of categories, and lots of “noisy examples”. The exception with which we are familiar with is the application of dropout to depth-estimation and segmentation [1], which considers uncertainty in a scenario much closer to reality.
> > >
> > > Secondly, WMT’17 EN-DE, WMT’14 EN-FR and LibrSpeech large scale and complex tasks. Indeed, they are larger and more complex than, for example ImageNet. WMT’17 EN-DE contains about 4 million sentence-pairs, the models are defined over 40K classes (BPE-tokens), the data has inherent multi-modality (multiple translations are valid).  WMT’14 EN-FR is about 40 million sentence pairs - an order-of-magnitude jump in scale. Our LibriSpeech data contains 1000 hours of speech data, the model is defined over 5K classes (sentence-piece tokens). The autoregressive nature of the models makes the hypothesis space vast, which poses interesting challenges. To put this in context - ImageNet has 1 million examples and 1K categories, and the hypothesis space is easily enumerable, as it coincides with the number of classes.
> > >
> > > Thirdly, Transformer-BIG and VGG-Transformer are large models which, or architectures very close to which, are currently used in production scenarios we are aware of.
> > >
> > > It is in this context that we claim to be working on models and datasets which are far larger and more complex than what is common on most Bayesian research, and which are not that far removed from industrial ASR/NMT systems with which we are intimately familiar.
> > >
> > > However, we accept that we may have missed the application of Bayesian uncertainty estimation to tasks in other areas which are as complex if not more so than the ones we consider here. If you could provide examples of such tasks and applications, we would be happy to moderate our claims.
> > >
> > > [1] What Uncertainties Do We Need in Bayesian Deep Learning for Computer Vision? Gal & Kendall, 2017

---

### Official Review · AnonReviewer3 · 2020-10-29
**Official Blind Review #3**

**Rating:** 7
**Confidence:** 4

**Review:**

After rebuttal: I would like to thank the authors for paying attention to the comments and providing additional experiments and results.
I updated my score.

==================================
This paper investigates uncertainty estimation for autoregressive structured prediction models. The authors explored several metrics for computing uncertainty in structured neural network based models, such as expected pair-wise KL-divergence, mutual information, and reverse mutual information (RMI). The authors demonstrated the efficiency of the proposed RMI using machine translation and speech recognition models for error detection and out-of-distribution detection.

Overall this paper is clearly written and investigates highly relevant topic. However, some clarifications are needed. First, I find the name of the paper a bit misleading, there are plenty of structured prediction models, and the authors investigated only neural based and autoregressive ones.

Second, I know the authors mentioned that they did not compare to prior work since it did not consider auto-regressive models, however, I feel that a comparison is still needed. There are a few papers (some of which the authors already mentioned), that suggest different ways to estimate uncertainties in AST and MT. I would expect to see a comparison to some of these methods.

Comments to the authors:

1) In "Practical Considerations" there is a question mark, probably a missing citation?

2) The improvements in Table 1 seems pretty marginal. Can the authors report the variance of the estimator too?

3) "The results also show that uncertainty-based rejection works better for ASR for NMT." → "The results also show that uncertainty-based rejection works better for ASR than NMT."

4) "A better, but more expensive, approach to assess uncertainty estimates in NMT is whether they correlate well with human assessment of translation quality." → did the authors try to use subjective evaluations?

5) "Three OOD datasets are considered, each covering a different form of domain shift. First, LibriSpeech test-other (LTO)..." → I find this definition of OOD a bit problematic. LTO is still part of librispeech, so I would not consider that as OOD.

6) "Notably, it was found that token-level Bayesian model averaging consistently yields both marginally better predictive performance and more robust estimates of uncertainty." → One reason for that is maybe since the authors are normalizing their probabilities at the token level and not at the sequence level? did the authors investigate such models that are normalizing at the seq level such as: [1].

I'm willing to increase my score if the above questions will be answered

[1] Collobert, Ronan, Christian Puhrsch, and Gabriel Synnaeve. "Wav2letter: an end-to-end convnet-based speech recognition system." arXiv preprint arXiv:1609.03193 (2016).

---

> ### Author Response · Authors · 2020-11-13
> **Response to Reviewer 3**
>
> Thank you for your comments!
>
> 1. Regarding the name:
>
> We understand where you're coming from - this generated a lot of internal debate. However, we settled on "Uncertainty in Structured Prediction" rather than "Uncertainty in Autoregressive Prediction" because we felt that we were tackling the most general model which made the fewest conditional independence assumptions. If we were to consider non-autoregressive models (such as CTC, HMM, etc...), for example, which make stronger conditional-independence assumptions, then the proposed methods would be as applicable, and there would also be no need for  Monte-Carlo approximations.
>
> 2. Baselines
>
> We understand the desire for additional benchmarks, and are happy to provide ones which we feel are appropriate. Let us clarify what we feel is appropriate:
>
> Ensemble based uncertainty estimation is both *general* (not task specific) and *unsupervised*, which means we have no supervision regarding where errors are and what is in-domain or out-of-domain. This is a very desirable setup. We have made comparisons with prior heuristic approaches to unsupervised ensemble-based uncertainty estimation in NMT (and have adapted those methods to ASR), the comparison is provided in appendix H table 15. We show that the proposed methods outperform prior heuristics. We will move these results to the main paper, as space now allows for this.
>
> Regarding prior work in confidence-score estimation in ASR - all of it is basically *supervised* - here either a separate classifier is built on top of an ASR system or some kind of mapping is used to transform the acoustic-model likelihoods into probability of insertions/substitutions at each work position. The ground truth error labels are obtained by aligning the true transcription on a validation set to the ASR system’s hypotheses and training relative to that supervision. This has advantages - that this explicitly targets error detection. However, this additional model is also sensitive to various domain shift, does not allow separation of uncertainty into total/knowledge, and requires supervision which itself can be noisy. Thus, we do not believe that a comparison of general, unsupervised ensemble-based uncertainty estimation to supervised methods is appropriate.
>
> However, the basis of these supervised systems are ASR  system likelihoods, to which *we are already comparing* every time we consider $\mathcal{\hat H}_{\text{S-IW}}^{(1)}$ from the top beam search hypothesis. We will make this clear in the text using the additional page made available.
>
> REGARDING COMMENTS:
> 1. Yes, broken citation. Will fix.
> 2. +/- 2*sigma is reported in the ablation study in figures 1 and 2 in the appendices.  Here we leave one or more elements of the ensemble out and compute mean/variance across possible sub-ensembles. This is the best we can do for providing confidence bounds for ensemble performances, as we have limited compute to train multiple large ensembles.
> 3. Thanks for finding, will fix.
> 4. Regarding human-eval - We haven't done this, but it is something we are very much considering doing as part of a future shared task.
> 5. Regarding LibriSpeech - You are correct in that LTO is also a part of LibriSpeech, and perhaps we should not strictly refer to it as OOD. This really boils down to how one defines OOD/anomalies. To our understanding, LTO is a dataset collected from samples that were challenging to correctly transcribe. This may have been due to a higher level of minor anomalies and datashifts. AS LTO is a dataset on which, for whatever reason, the ASR system makes more errors, and we would like the model to be able to distinguish between data on which it is likely to make more errors from data on which it makes fewer errors. We felt the cleaner comparison would have been to treat it as anomaly detection, rather than sequence-error detection (lumping LTC and LTO into one big dataset).
> 6. Global vs Local Norm - The locally normalized nature of autoregressive models may indeed be part of the problem. This may also be a consequence of the fact that the model’s representation of context is imperfect. As discussed above, we haven’t considered non-autoregressive sequence-trained globally normalized state-space models, and it would definitely be interesting to do so! Unfortunately, setting this up and training an ensemble would require more time than is available in the rebuttal.

---

> ### Author Response · Authors · 2020-11-20
> **Regarding updated manuscript**
>
> Duplicating for clarity - Do you find the extra baselines to other unsupervised methods sufficient?

---

> > ### Comment · AnonReviewer3 · 2020-11-24
> > **Updated reivew**
> >
> > I would like to thanks the authors for the additional experiments and I appreciate the authors' hard work.
> > After reading the other reviews, authors' response and the updated manuscript, I find this paper very interesting and important. The additional experiments and results are sufficient and make the authors claim significantly stronger.
> >
> > I agree w. reviewer 1 that adding experiments on other structured tasks such as image segmentation could make this submission stronger and more general, however after the authors changed the name of this paper to better match its content and experiments, such experiments are unnecessary.
> >
> > I will update my review and increase my score.

---

### Official Review · AnonReviewer2 · 2020-11-02
**AnonReviewer2**

**Rating:** 7
**Confidence:** 3

**Review:**

* **Summary**:
This work introduce rigorous information-theoretic measures for structured prediction tasks. It proposed metrics for both `"total uncertainty" (entropy) and "knowledge uncertainty" (MI, EPKL and RMI) on the sequence level, introduced efficient Monte-Carlo approaches to estimate them in practice. Finally, author conducted thorough experiment to evaluate the effect of ensemble strategy and choice of metric for error detection and OOD detection in ASR and NMT.

* **Strength and Weakness**:
 * (Strength) Uncertainty quantification in structured prediction is an important but less explored topic. This work provided a much needed, information-theoretic framework to both conceptually quantify and empirically compute uncertainty measures for different sources of uncertainty.
 * (Strength) Thorough experiment on two tasks (ASR and NMT), where authors designed experiments to quantitatively measure uncertainty quality (Sequence-level Error Detection and OOD Detection) under different metrics and ensemble strategies.
 * (Weakness) The experiment conducted are mostly empirical where the ground truth is not necessarily known. Given the theoretical nature of this work, it might be good to conduct simulation study under known truth to examine the estimation quality of different metrics.

* **Recommendation**: Acceptance. I believe this work provided a nice theoretical framing of different uncertainty measures for an important and less-explored area (structured prediction). It also conducted thorough empirical investigation showing the relative merit of different modeling and measure choices on two standard structure prediction task. The information contained in this paper should be of sufficient interest to ICLR community.

* **Minor Comments**:
   * Missing reference on bottom of page 4: "rarely used in practice as it yields poor predictive performance ?."
   * Middle of page 6: "Rejection curves are summarised using the Prediction Rejection Ratio (PRR)". Make sure to mention PRR is introduced in detail in Appendix D.
    * Top of page 3 "As will be shown later, RMI is particularly attractive ..". It might be helpful to point to the section where the "attractiveness" of RMI is illustrated (Equation 12-13?).

---

> ### Author Response · Authors · 2020-11-13
> **Response to Reviewer 2**
>
> Thank you for your comments!
>
> We agree  and would have also liked to run a small synthetic task which allows us to do analysis in a scenario which we fully control. However, it is difficult to construct a synthetic task where it is possible to BOTH obtain the ground truth uncertainty, and which is still meaningfully complex. We would be happy to consider any suggestions you have!
>
> Typos are fixed and we will shortly update the submission.

---

### Author Response · Authors · 2020-11-20
**Updated Manuscript**

Dear Reviewers,

We have updated our manuscript, and hopefully have addressed your concerns more fully.

Major modification:

1. We have added discussion of unsupervised vs. supervised uncertainty estimation to the introduction. Hopefully this puts into better context what the appropriate baselines are.

2. We have moved the comparison to other, 'heuristic' ensemble-based uncertainty estimation approaches in NMT to the end of the experimental section, and have discussed what we see are the primary differences between our measures of uncertainty and these heuristic ones. The appendix (appendix I) has now been expanded to show exactly what we derive and calculate.
     @ Reviewer 3 - Do you find these results and added baselines sufficient?

3. We have added a section to the appendix (appendix H) which evaluates the effects of adding/removing length normalisation. We stated that this was important, but didn't evaluate this. Now we have, and it is clear that length norm is indeed important. This also means that the previous appendix H is now appendix I, and the current appendix H is new.

Minor modification:

1. We have fixed the typos, broken reference, and clarified notation (@ reviewer 1).

Finally, regarding the discussion of the title - upon uploading the update we noticed that we CAN change the title. If all reviewers are happy to allow us to do this, we would be happy to modify our title and moderate it to something like, for example, "Uncertainty Estimation in Autoregressive Structured Prediction".

---

### Author Response · Authors · 2020-11-23
**Updated Title**

Dear Reviewers,

As per your comments, we have modified the title from "Uncertainty in Structured Prediction" to "Uncertainty Estimation in Autoregressive Structured Prediction" in order to align the title better with the content of the work.

Many thanks for all your comments,
Authors

---

### Decision · Program_Chairs · 2021-01-07
**Final Decision**

**Decision:**

Accept (Poster)

**Comment:**

This paper proposes information-theoretic quantification of epistemic uncertainty in autoregressive models.

This is a difficult problem that receives much less attention than the unstructured case. The paper is well-written, contributes novel and tractable-to-estimate measures which are analysed formally and empirically with convincing experiments on ASR and NMT.

The reviewers and myself are overall pleased by this submission. The discussion phase went well and most concerns have been resolved.